

**Characteristics of Vertical Air Motion in Convective Clouds**
**Jing Yang[1], Zhien Wang[1], Andrew J. Heymsfield[2] and Jeffrey R. French[1]**
[1] {Department of Atmospheric Science, University of Wyoming, Laramie, WY}
[2] {National Center for Atmospheric Research, Boulder, CO}
Correspondence to: Zhien Wang (zwang@uwyo.edu)
**Abstract**
The vertical velocity and air mass flux in convective clouds are statistically analyzed using
aircraft in-situ data collected from three field campaigns: High-Plains Cumulus (HiCu)
conducted over the mid-latitude High Plains, COnvective Precipitation Experiment (COPE)
conducted in a mid-latitude coastal area, and Ice in Clouds Experiment-Tropical (ICE-T),
conducted over a tropical ocean. This study yields the following results. (1) Small-scale updrafts
and downdrafts (< 500 m in diameter) are frequently observed in the three field campaigns, and
they make important contributions to the total air mass flux. (2) The probability density functions
(PDFs) of the vertical velocity are exponentially distributed. For updrafts, the PDFs of the
vertical velocity are broader in ICE-T and COPE than in HiCu; for downdrafts, the PDFs of the
vertical velocity are broader in HiCu and COPE than in ICE-T. (3) Vertical velocity profiles



show that updrafts are stronger in ICE-T and COPE than in HiCu, and downdrafts are stronger in
HiCu and COPE than in ICE-T. (4) The PDFs of the air mass flux are exponentially distributed
as well. The maximum air mass flux in updrafts is of the order $10^4$ kg m$^{-1}$ s$^{-1}$. The air mass flux
in the downdrafts is typically a few times smaller in magnitude than that in the updrafts.
**1.    Introduction**
Convective clouds are an important component of the global energy balance and water cycle
because they dynamically couple the planetary boundary layer to the free troposphere through
vertical heat, moisture and mass transport (Arakawa, 2004; Heymsfield et al., 2010; Wang and
Geerts, 2013). The vertical velocity determines the vertical transport of cloud condensate, the
cloud top height and the detrainment into anvils, which further impact the radiative balance (Del
Genio et al., 2005). Vertical velocity also has significant impact on the aerosol activation, droplet
condensation and ice nucleation in convective clouds, which control the cloud life cycle and
precipitation efficiency.
In order to reasonably simulate convective clouds, the vertical air velocity must be parameterized
reliably in numerical weather prediction models (NWPMs) and global circulation models (GCMs)
(Donner et al., 2001; Tonttila et al., 2011; Wang and Zhang, 2014). However, the complexity of
the vertical velocity structure in convective clouds makes the parameterization non-
straightforward (Wang and Zhang, 2014). Observations show that in most of the convective
clouds the vertical velocity is highly variable, and consequently the detailed structure of
convection cannot be resolved in many models (Kollias et al., 2001; Tonttila et al., 2011).



Additionally, using the same parameterization of vertical velocity for different grid resolutions
may result in different cloud and precipitation properties (Khairoutdinov et al., 2009).
Furthermore, poorly parameterized vertical velocity may result in large uncertainties in the
microphysics; for instance, the cloud droplet concentration may be underestimated due to
unresolved vertical velocity (Ivanova and Leighton, 2008). Vertical velocity simulated by
models with horizontal resolutions down to a few hundred meters may be more realistic (e.g. Wu
et al., 2009), but more observations are needed to evaluate this suggestion.
Aircraft in-situ measurement has been the most reliable tool enabling us to understand the
vertical velocity in convective clouds and to develop the parameterizations for models. Early
studies (e.g. Byers and Braham, 1949; Schmeter, 1969) observed strong updrafts and downdrafts
in convective clouds; however, their results have a large uncertainty, because the aircrafts were
not equipped with inertial navigation systems (LeMone and Zipser, 1980). In 1974, the Global
Atmospheric Research Program (GARP) Atlantic Tropical Experiment (GATE) was conducted
off the west coast of Africa, focusing on tropical maritime convections (Houze, 1981). A series
of findings based on the aircraft data collected from the project was reported. For example, the
accumulated probability density functions (PDFs) of vertical velocity and diameter of the
convective cores are lognormal distributed. The updrafts and downdrafts in GATE (tropical
maritime clouds) were only one half to one third as strong as those observed in the Thunderstorm
Project (continental clouds) (LeMone and Zipser, 1980; Houze, 1981). These findings stimulated
later statistical studies of the vertical velocity in convective clouds. Jorgensen et al. (1985) found
that the accumulated PDFs of vertical velocity in intense hurricanes were also lognormal
distributed and the strength was similar to that in GATE, but the diameter of the convective
region was larger. Studies of the convective clouds over Taiwan (Jorgensen and LeMone, 1989)



and Australia (Lucas et al., 1994) showed a magnitude of vertical velocity similar to that in
GATE. Although the results from the Thunderstorm Project are suspect, the significantly
stronger drafts reveal the possible difference between continental and tropical maritime
convective clouds. Lucas et al. (1994) suggested that the water loading and entrainment strongly
reduce the strength of updrafts in maritime convections. However, this underestimation of the
updraft intensity may be also due to the sampling issues, e.g. penetrations were made outside the
strongest cores (Heymsfield et al., 2010).
There are a few more recent aircraft measurements (e.g. Igau et al, 1999; Anderson et al., 2005),
but the data are still inadequate to fully characterize the vertical velocity in convective clouds. In
most of these earlier papers, the defined draft or draft core required a diameter no smaller than
500 m; this threshold excluded many narrow drafts with strong vertical velocity and air mass
flux. In addition, the earlier studies used 1-Hz resolution data, which can resolve only the vertical
velocity structures larger than a few hundred meters, but the narrow drafts may be important to
the total air mass flux exchange and cloud evolution. Furthermore, previous aircraft observations
for continental convective clouds were based only on the Thunderstorm Project; thus, new data
are needed to study the difference between continental and maritime convections.
Remote sensing by means of, for example, wind profilers and radars is another technique which
has often been used in recent years for studying the vertical velocity in convective clouds (e.g.
Kollias et al., 2001; Hogan et al., 2009; Schumacher et al., 2015). Using profiler data, May and
Rajopadhyaya (1999) analyzed the vertical velocity in deep convections near Darwin, Australia.
They observed that the updraft intensified with height and that the maximum vertical velocity
was greater than 15 m s$^{-1}$. Heymsfield et al. (2010) studied the vertical velocity in deep



convection using an airborne nadir-viewing radar. Strong updrafts were observed over both
continental and ocean areas, with the peak vertical velocity exceeding 15 m s$^{-1}$ in most of the
cases and exceeding 30 m s$^{-1}$ in a few cases. Zipser et al. (2006) used satellite measurements to
find the most intense thunderstorms around the world; they applied a threshold updraft velocity
greater than 25 m s$^{-1}$ to identify intense convection. Remote sensing has the advantage of being
able to measure the vertically velocity at different heights simultaneously (Tonttila et al., 2011).
However, remote sensing measurements are not as accurate as aircraft measurements, because
many assumptions are needed to account for the contribution of particle fall speed in the
observed Doppler velocity in order to ultimately estimate air velocity. In addition, ground-based
radars can rarely provide good measurements over oceans, and airborne cloud radars often suffer
from the attenuation and non-Rayleigh scattering in convective clouds. Therefore, in-situ
measurements are still necessary in order to characterize the dynamics in convective clouds and
to develop parameterizations for models.
The present study provides aircraft data analysis of the updrafts and downdrafts in mid-latitude
continental, mid-latitude coastal and tropical maritime convective clouds using the fast-response
in-situ measurements collected from three field campaigns: the High-Plains Cumulus (HiCu), the
COnvective Precipitation Experiment (COPE) and the Ice in Clouds Experiment-Tropical (ICE-
T). All the clouds formed in isolation, but some of them merged as they evolved. Statistics of the
vertical velocity and air mass flux are provided. The Wyoming Cloud Radar (WCR), onboard the
aircraft, is used to identify the cloud top height, and high frequency (25-Hz) in-situ
measurements of vertical velocity are used to generate the statistics. Section 2 describes the
datasets and wind measuring systems. Section 3 presents the analysis method. Section 4 shows
the results, and conclusions are given in Section 5.




## 2.    Dataset and instruments

### 2.1    Dataset

The data used in the present study were collected from three field campaigns: HiCu, COPE and
ICE-T. Vigorous convective clouds were penetrated during the three field campaigns, including
mid-latitude continental, mid-latitude coastal, and tropical maritime convective clouds. These
penetrations provide good quality measurements for studying the microphysics and dynamics in
the convective clouds, as well as the interactions between the clouds and the ambient air. The
locations of the three field campaigns are shown in Fig. 1. Information regarding the penetrations
used in this study is summarized in Table 1.
The HiCu project was conducted mainly in Arizona and Wyoming (Fig. 1) from 18 July to 05
August 2002 and from 07 July to 31 August 2003 to investigate the microphysics and dynamics
in convective clouds over mid-latitude High Plains. The University of Wyoming King Air
(UWKA) was operated as the platform. In 2002 and 2003, 10 and 30 research flights were made,
respectively. In this study, the 2002 HiCu and 2003 HiCu are analyzed together because they
were both conducted over the High Plains and the sample size of 2002 HiCu is relatively small.
Fast-response in-situ instruments and the Wyoming Cloud Radar (WCR, Wang et al., 2012) were
operated during the field campaign to measure the ambient environment, cloud dynamics and
microphysics as well as two-dimensional (2D) cloud structure. As shown in Table 1, penetrations
in HiCu were made between 2 km and 10 km MSL. The sample size is relatively good below 8
km and relatively small above 8 km. The aircraft flew about 2000 km in clouds. In-situ



measurements and WCR worked well in these flights; however, the upward-pointing radar was
operated in less than half of the research flights, and thus only a sub-set of the cloud tops can be
estimated. Fig. 2a(1–3) shows an example of the clouds sampled in HiCu, including WCR
reflectivity, Doppler velocity and 25-Hz in-situ measurement of the vertical velocity. In HiCu,
both developing and mature convective clouds were penetrated; some penetrations were near
cloud top, while most of them were more than 1 km below cloud top. From the Doppler velocity
and the in-situ vertical velocity, we can see that, in both the developing and mature cloud, strong
updrafts and downdrafts were observed, and multiple updrafts and downdrafts existed in the
same cloud.
The COPE project was conducted from 03 July to 21 August, 2013 in Southwest England (Fig.
1). The UWKA was used to study the microphysics and entrainment in mid-latitude coastal
convective clouds (Leon et al., 2015). Seventeen research flights were conducted; penetrations
focused on regions near cloud top, which is verified based on the radar reflectivity from the
onboard WCR. Since COPE was conducted in a coastal area, the convection initiation
mechanism is different from that over a purely continental or ocean area. In addition, although
the ambient air mainly came from the ocean, continental aerosols might be brought into clouds,
since many of the convective clouds formed within the boundary layer, which further affects the
microphysics and dynamics in the clouds. The measurements made in COPE include temperature,
vertical velocity, liquid water content, and particle concentration and size distributions. The
WCR provided excellent measurements of reflectivity and Doppler velocity. The downward
Wyoming Cloud Lidar (WCL) was operated to investigate the liquid (or ice) dominated clouds.
Between 0 km and 6 km, about 800 penetrations were made. Flight distance in cloud totaled
about 1000 km. The sample sizes are relatively good between 2 km and 6 km, but relatively



small between 0 km and 2 km. Examples of the penetrations are given in Fig. 2b(1–3). COPE has
fewer penetrations than HiCu, and most of the penetrations are near the cloud top. Fig. 2b(2)
reveals relatively simple structures of the updrafts and downdrafts in COPE compared to HiCu,
but as shown by the 25-Hz in-situ vertical velocity measurement in Fig. 2b(3), there are still
many complicated fine structures in the vertical velocity distribution.
The ICE-T project was conducted from July 1 to July 30, 2011 near St. Croix, U.S. Virgin
Islands (Fig. 1), with state-of-the-art airborne in situ and remote sensing instrumentations, with
the aim of studying the role of ice generation in tropical maritime convective clouds. The
NSF/NCAR C-130 aircraft was used during ICE-T to penetrate convective clouds over the
Caribbean Sea. Thirteen C-130 research flights were conducted during the field campaign, with
vigorous convective clouds penetrated. In-situ measurements from ICE-T include the liquid and
total condensed water contents, temperatures, vertical velocities, and cloud and precipitating
particle concentrations and size distributions. The WCR was operated on seven research flights
to measure the 2D reflectivity and Doppler velocity fields. The aircraft flew more than 1500 km
in clouds, and more than 650 cloud penetrations were made between 0 km and 8 km. The sample
sizes are good except between 2 km and 4 km (Table 1). Examples of the penetrations are shown
in Fig. 2c(1–3). During ICE-T, clouds in different stages were penetrated, including developing,
mature and dissipating, some near cloud top and some considerably below cloud top. Strong
updrafts were observed in the developing and mature clouds, but the downdrafts in ICE-T are
typically weaker than those in HiCu and COPE. The vertical velocity structures are complicated,
as confirmed by both the Doppler velocity and the 25-Hz in-situ measurement. Weak updrafts
and downdrafts were also observed in the dissipating clouds.




## 2.2 Wind measuring system

On both C-130 and UWKA, A Radome Five-Hole Gust Probe is installed for three-dimensional
(3D) wind measurement**.** A Radome Five-Hole Gust Probe is an aircraft radome probe with five
pressure ports installed in a "cross" pattern. Relative wind components (e.g. true air speed and
flow angles) are sensed by a combination of differential pressure sensors attached to the five
holes (Wendisch and Brenguier, 2013). Detailed calculation of relative wind components is
described in Kroonenberg et al. (2008) and Wendisch and Brenguier (2013). The time response
and the accuracy of the pressure sensors is about 25 Hz and 0.1 mb. The 3D wind vectors can be
derived by taking out the aircraft motions from the relative wind measurement. On both C-130
and UWKA, the aircraft motion is monitored by a Honeywell Laseref SM Inertial Reference
System (IRS), with an accuracy of 0.15 m s$^{-1}$ for vertical motion. Global Positioning System
(GPS) was applied to remove the drift errors in the IRS position in all the three field campaigns
(Khelif et al., 1998). The final vertical wind velocity product has an accuracy of about ±0.2 m s$^{-1}$,
and a time response of 25 Hz. This uncertainty (±0.2 m s$^{-1}$) is a mean bias. For each output, the
uncertainty is related to the true air speed, aircraft pitch angle, roll angle and ambient conditions.
Therefore, the random error varies and could be larger than the mean bias. More information
about the wind measurement on C-130 and UWKA can be found on the C-130 Investigator
Handbook (available on https://www.eol.ucar.edu/content/c-130-investigator-handbook) and
UWKA Investigator Handbook (available on
http://www.atmos.uwyo.edu/uwka/users/KA_InstList.pdf)





## 3. Analysis method

### 3.1 Identifying cloud using in-situ measurements

The Particle Measuring Systems (PMS) Two-Dimensional Cloud (2D-C) Probe and the Forward
Scattering Spectrometer Probe (FSSP) are often used to characterize cloud microphysics (e.g.
Anderson et al., 2004), although different thresholds of 2D-C and FSSP concentrations are
usually used to identify the edge of a cloud. In this paper, we also use FSSP and 2D-C probes to
find the cloud edges. In order to find a reasonable threshold for identifying cloudy air, we first
use the WCR reflectivity to identify the clouds and the cloud-free atmosphere; for those regions
we then plot the particle concentrations measured by FSSP and 2D-C in order to determine the
reasonable thresholds, and we apply the thresholds of particle concentrations to all the research
flights without WCR.
To identify clouds using WCR, the six effective range gates nearest to the flight level (three
above and three below) are chosen in each beam. Any beam in which the minimum reflectivity at
the six gates exceeds the noise level[1] is identified as in cloud.
Fig. 3 shows the occurrence distribution as a function of the particle concentrations measured by
FSSP versus the concentrations of the particles $\geq 50$ μm in diameter measured by 2D-C in the
clouds identified by WCR reflectivity. From the figure, we can see that the FSSP concentration
ranges from 0.01 cm$^{-3}$ to 1000 cm$^{-3}$, and the 2D-C concentration ranges from 0.1 L$^{-1}$ to 10000 L$^{-1}$.
Generally, shallow clouds have relatively higher concentrations of small particles and lower

---

[1] Based on the reflectivity measured in cloud-free air, the noise level of WCR reflectivity is -32 dBZ at a range of 500 m and -28 dBZ at a range of 1000 m. In this study, we choose -30 dBZ as the threshold to identify cloud. This threshold is examined for all three field campaigns.





concentration of particles larger than 50 μm. In deeper convective clouds, high concentrations
can be seen for both small and large particles. The FSSP concentrations in cloud-free air are
found to be 2 cm$^{-3}$ at most, and the FSSP concentrations measured below the lifting condensation
level (LCL), where precipitating particles dominated, are lower than 2 cm$^{-3}$, as well. Therefore, 2
cm$^{-3}$ is selected as the concentration threshold to identify clouds based on the FSSP
measurements, as shown by the dashed line in Fig. 3. However, in some clouds (e.g. pure ice
clouds), the FSSP concentration could be lower than 2 cm$^{-3}$, and 2D-C concentrations are needed
to identify these cold clouds. We chose a 1 L$^{-1}$ 2D-C concentration for particles ≥ 50 μm as the
second threshold to identify cloud, as shown by the dotted line in Fig. 3. In order to avoid
precipitating regions (below the LCL calculated from soundings), the second threshold is only
applied to penetrations at temperatures colder than 0 ℃; thus the cloud is defined as FSSP
concentration ≥ 2 cm$^{-3}$ or 2D-C concentration ≥ 1 L$^{-1}$. At temperatures warmer than 0 ℃, the
FSSP concentrations in most of the convective clouds are higher than 2 cm$^{-3}$, so only the first
threshold is used.
Once a cloud is identified, the penetration details can be calculated, including the flight length,
the flight height, the cloud top height if WCR is available, and the penetration diameter. The
penetration diameter is calculated as the distance between the entrance and exit of a penetration.
In order to reject whirl penetrations and penetrations with significant turns, we require that the
diameter of a penetration be at least 90% of the flight length. The penetration diameter can
generally reveal the scale of a cloud, but since the aircraft may not penetrate exactly through the
center of a cloud, the actual cloud diameter may be larger than the penetration diameter. Based
on WCR reflectivity images, there are no isolated convective clouds larger than 20 km in





diameter. There are a few penetrations longer than 20 km, but these clouds are more like
mesoscale convective systems (MCS), and so they are excluded from this study.

**3.2    Defining updraft and downdraft**
In previous studies of the vertical velocity based on in-situ measurements, the updraft and
downdraft are often defined as an ascending or subsiding air parcel with the vertical velocity
continuously $\geq 0$ m s$^{-1}$ in magnitude and $\geq 500$ m in diameter (e.g. LeMone and Zipser, 1980;
Jorgensen and LeMone, 1989; Lucas et al., 1994; Igau et al., 1999). In this study, we use a
vertical velocity threshold of 0.2 m s$^{-1}$, that is, the draft has a vertical velocity continuously $\geq 0.2$
m s$^{-1}$ in magnitude, because $\pm 0.2$ m s$^{-1}$ is the accuracy of the instrument. Any very narrow and
weak portion (diameter < 10 m and maximum vertical velocity < 0.2 m s$^{-1}$ in magnitude)
between two relatively strong portions is ignored, and the two strong portions are considered as
one draft.
The diameter threshold (500 m) is not used in this paper, because drafts narrower than 500 m
frequently occur and they make important contributions to the total air mass flux in the
atmosphere and therefore they are necessarily to be considered in model simulations. Fig. 4
shows the PDFs of the diameters of all the updrafts and downdrafts sampled in HiCu, COPE and
ICE-T. In all the panels, the diameters are exponentially distributed, the PDFs can be fitted using
$$f = \alpha \cdot |x|^{\beta} \cdot \exp(\gamma|x|) \qquad (1)$$



where $f$ is the frequency and $x$ is the diameter. The coefficients $\alpha$, $\beta$ and $\gamma$ for each PDF is shown
in each panel. This function will also be used to fit the PDFs of vertical velocity and air mass
flux in the following analyses. Generally, as seen in Fig 4, the PDFs broaden with height
increases for the three field campaigns; this is consistent with previous findings (LeMone and
Zipser, 1980). The diameters of the updrafts are smaller in COPE compared to those sampled in
HiCu and ICE-T, possibly because most of the penetrations are near cloud top. The diameters of
the downdrafts are relatively small in HiCu. ICE-T has the most drafts with diameters exceeding
100 m, and the average diameters in ICE-T for both updrafts and downdrafts are the largest. As
shown in Fig. 4, many narrow drafts are observed. More than 85%, 90% and 74% of the updrafts
are narrower than 500 m (dotted lines) in HiCu, COPE and ICE-T, respectively, and more than
90% of the downdrafts in all three field campaigns are narrower than 500 m. A threshold of 500
m in diameter would exclude many small-scale drafts, therefore, in this study all the drafts
broader than 50 m (dashed lines) are included. The drafts narrower than 50 m are excluded
because most of them are turbulences and they can hardly be resolved in models.
Fig. 5a shows the occurrence distributions as a function of the mean vertical velocity versus the
diameter of the drafts with the vertical velocity continuously $\geq 0.2$ m s$^{-1}$ in magnitude. From the
figure, it is noted that many drafts narrower than 500 m have quite strong vertical velocities. The
maximum mean vertical velocity of these narrow drafts can reach 8 m s$^{-1}$, and the minimum
mean vertical velocity in the downdrafts is $-6$ m s$^{-1}$. With such strong mean vertical velocity,
narrow drafts could contribute noticeably to the total air mass flux. Fig. 5b presents the
occurrence distributions as a function of the air mass flux versus the diameter of the drafts. The
air mass flux is calculated as $\bar{\rho}\bar{w}D$ (LeMone and Zipser, 1980), where $\bar{\rho}$ is the mean air density
at the measurement temperature, $\bar{w}$ is the mean vertical velocity and $D$ is the diameter of each





draft. Fig. 5b shows that the air mass flux in many drafts narrower than 500 m is actually larger
than that in some of the broader drafts. The maximum value for these narrow updrafts reaches
4000 kg m$^{-1}$ s$^{-1}$, and the minimum value for the downdrafts reaches –3000 kg m$^{-1}$ s$^{-1}$. The
normalized accumulated flux (red curves) reveals that the drafts narrower than 500 m (dotted
horizontal lines) make very significant contributions to the total air mass flux. Calculations
indicate that the updrafts narrower than 500 m contribute 20%–35% of the total upward flux, and
that the downdrafts narrower than 500 m contribute 50%–65% of the total downward air mass
flux. Drafts narrower than 50 m (dashed horizontal lines), which are excluded in this paper,
contributes less than 5% of the total air mass flux.
In this study, we delineate three different groups of updraft and downdraft using three thresholds
of air mass flux: 10 kg m$^{-1}$ s$^{-1}$, 100 kg m$^{-1}$ s$^{-1}$ and 500 kg m$^{-1}$ s$^{-1}$ in magnitude., The air mass flux
is used here to delineate the draft intensity because (1) air mass flux contains the information of
both vertical velocity and draft size; (2) air mass flux can reveal the vertical mass transport
through convections; and (3) air mass flux is an important component in cumulus and convection
parameterizations (e.g. Tiedtke, 1989; Bechtold et al., 2001). The first designated group, the
"weak draft," with air mass flux 10–100 kg m$^{-1}$ s$^{-1}$ in magnitude, contributes 10% of the total
upward air mass flux and 10% of the total downward air mass flux. The "moderate draft," with
air mass flux 100–500 kg m$^{-1}$ s$^{-1}$ in magnitude, contributes 25% of the total upward air mass flux
and 40% of the total downward air mass flux. The "strong draft," where the air mass flux ≥ 500
kg m$^{-1}$ s$^{-1}$ in magnitude contributes 60% of the total upward air mass flux and 20% of the total
downward air mass flux. Drafts weaker than 10 kg m$^{-1}$ s$^{-1}$ are not analyzed because they are too
weak and most of them are very narrow and can hardly be resolved in models (Fig. 5b). The
numbers of weak, moderate and strong updrafts and downdrafts sampled at 0–2 km, 2–4 km, 4–6



km, 6–8 km and 8–10 km MSL are shown in Table 2. Generally, weak and moderate drafts are
more often observed than strong drafts. At most of the height ranges, more updrafts are observed
than downdrafts.
Some researchers have defined a "draft core" by selecting the strongest portion in a draft. For
example, LeMone and Zipser (1980) define an updraft core as an ascending air motion with
vertical velocity continuously $\geq 1$ m s$^{-1}$ and diameter $\geq 500$ m. This definition of a "draft core" is
followed in a few more recent studies (e.g. Jorgensen and LeMone, 1989; Lucas et al., 1994;
Igau et al., 1999). We too analyzed the vertical air motion characteristics in the stronger portion
of the drafts considered here. However, we found that in many updrafts the strong portion where
the vertical velocity is continuously $\geq 1$ m s$^{-1}$ dominates and contributes 80% of the total air
mass flux, so the statistics of the vertical air motion characteristics in the stronger portion are
very similar to those in the draft as a whole. Therefore, the present study focuses on "drafts" in
which both weak and strong portions are included.

**4.    Results**
**4.1    Significance of drafts in different strengths**
From the analysis above, we note that relatively small and weak updrafts are frequently observed
in convective clouds. In this section, we provide further evidence to show the importance of the
relatively weak updrafts in terms of air mass flux.



Fig. 6a shows the average number of updrafts as a function air mass flux observed in the three
field campaigns. The solid, dashed and dotted lines represent the penetrations with different
diameters. As shown in Fig. 6a, weak and moderate updrafts are more often observed than strong
updrafts, and the numbers of updrafts are higher in longer penetrations. Since this is an average
result, the number of updrafts could be smaller than 1 (e.g. many narrow penetrations do not
have strong updrafts). Fig.6b is similar to Fig. 6a but shows the occurrence frequency of updrafts
with different air mass fluxes (i.e. the vertical axis in Fig. 6a is normalized). For the penetrations
< 1 km, many of the clouds only have weak or moderate updrafts, and strong updrafts are rarely
observed. For penetrations of 1−10 km, the frequency of strong updrafts increases and the
frequency of weak and moderate updrafts decreases. For even longer penetrations (>10 km),
however, the frequency of weak updrafts increases again, indicating the increasing importance of
weak updrafts.
Fig. 7 shows the average percentile contributions to the total upward air mass flux by the three
different groups of updrafts as a function of penetration diameter. In Fig. 7a, all the penetrations
are included. Since many narrow clouds have no strong updrafts in terms of air mass flux, the
total air mass flux in these narrow clouds is mostly contributed by weak (red bar) and moderate
(green bar) drafts. These narrow clouds may have a high vertical velocity but small air mass flux.
As the diameter increases to 4 km, the contributions to total air mass flux from relatively weak
updrafts (red bar) decrease, while those from stronger updrafts (blue bar) increase. For a
penetration of 4 km, 80%−90% of the total upward mass flux is contributed by the strong
updrafts with air mass flux $\geq 500$ kg m$^{-1}$ s$^{-1}$. However, for the penetrations with diameter larger
than 4 km, the contribution from relatively weak updrafts increases, probably because more
weak updrafts exist in wider clouds (Fig. 6). This is more obvious in Fig. 7b, in which only the

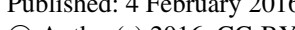


penetrations with at least one strong updraft are included. As the diameter increases from 400 m
to 20 km, the contribution from the weak and moderate updrafts (red bars and green bars)
increases from 2% to 20%. This suggests that as the cloud evolves and becomes broader (e.g.
mature or dissipating stage), the weak and moderate updrafts are also important and therefore
necessary to be considered in model simulations.

**4.2    PDFs of vertical velocity and air mass flux**
Fig. 8 shows the PDFs of the vertical velocity in the drafts sampled at 0–2 km, 2–4 km, 4–6 km
and higher than 6 km in the three field campaigns. Columns (a), (b) and (c) represent the drafts
with air mass flux $\geq$ 10 kg m$^{-1}$ s$^{-1}$, $\geq$ 100 kg m$^{-1}$ s$^{-1}$ and $\geq$ 500 kg m$^{-1}$ s$^{-1}$ in magnitude,
respectively; in other words, column (a) includes all the weak, moderate and strong of drafts,
column (b) includes moderate and strong updrafts, and column (c) includes strong updrafts only.
For statistical analysis, it is better to analyze different drafts together rather than separately. In all
the panels, the vertical velocities are exponentially distributed for both updrafts and downdrafts;
the PDFs can be fitted using Eq. (1). From Fig. 8 we can see that at 0–2 km, the PDFs for both
COPE and ICE-T are narrow; the updrafts in COPE are slightly stronger than those in ICE-T,
while the downdrafts are relatively weaker. At 2–4 km, stronger updrafts and broader PDFs are
observed in both COPE and ICE-T compared to those at 0–2 km; the maximum vertical velocity
is about 15 m s$^{-1}$. In COPE, the downdrafts are stronger than those in ICE-T, with the minimum
vertical velocity as low as –10 m s$^{-1}$. For HiCu, the PDFs of the vertical velocity at 2–4 km are
narrow, because the HiCu was conducted in the High Plains and the cloud bases are relatively
high. At 4–6 km, the updrafts become stronger and the PDFs become broader in all the three





field campaigns compared to those at lower levels, especially for COPE and ICE-T. Above 6 km,
the PDFs for the updraft become broader in HiCu while they slightly narrow in ICE-T compared
to those at 4–6 km. For the downdrafts, the PDFs broaden with height for all the three field
campaigns. Generally, the PDFs of the vertical velocity are similar for the three columns. The
main difference is found in the first bins of the vertical velocity ($0–2$ m s$^{-1}$ and $-2–0$ m s$^{-1}$):
highest for column (a), which includes all the drafts with air mass flux $\geq 10$ kg m$^{-1}$ s$^{-1}$ in
magnitude, lowest for column (c), which only includes the strong drafts with air mass flux $\geq 500$
kg m$^{-1}$ s$^{-1}$ in magnitude.
Generally, the updrafts are stronger in ICE-T or COPE (maritime or coastal convective clouds)
than in HiCu (pure continental convective clouds), an observation that differs from earlier studies
(e.g. LeMone and Zipser 1980), in which stronger drafts were observed in continental clouds.
This is probably because in the previous field campaigns over ocean (e.g. GATE), the aircraft did
not penetrate the strongest cores due to safety concerns. Compared to GATE, the PDFs of the
vertical velocity in ICE-T has a similar vertical dependence, broadening with height, but the
PDFs are broader in ICE-T than those in GATE, and the maximum vertical velocity (25 m s$^{-1}$) in
ICE-T is greater than that observed in GATE (15 m s$^{-1}$). In addition, convections in continental
areas other than the High Plains (e.g. Great Plains) may be different from those in HiCu.
Recently, Heymsfield et al. (2010) observed strong updrafts in both maritime and continental
convective  clouds: most exceed 15 m s$^{-1}$ and some exceed 30 m s$^{-1}$, but the measurements were
made for mature deep convection using airborne Doppler radar. More in-situ measurements are
needed to further evaluate the difference between maritime and continental convective clouds,
including both developing and mature stages.





There are a few possible explanations for the stronger updrafts observed in ICE-T and COPE
compared to those observed in HiCu. For example, the convective available potential energy
(CAPE) is larger in ICE-T than that in HiCu. Typically, the CAPE in ICE-T is greater than 2000
J kg$^{-1}$, and the CAPE in HiCu was less than 100 J kg$^{-1}$. However, CAPE in COPE is also low
(typically less than 100 J kg$^{-1}$), which cannot explain the relatively strong vertical velocity. The
strong vertical velocity in ICE-T and COPE maybe also be related to ice initiation. There are
many more millimeter drops in the convective clouds observed in ICE-T (Lawson et al., 2015)
and COPE (Leon et al., 2015) than that in HiCu; the millimeter drops can result in fast ice
initiation (Lawson et al., 2015), and the significant latent heat released during the ice initiation
process can strengthen the vertical velocity. In addition, high concentrations of millimeter drops
in ICE-T and COPE can result in the quick formation of graupel and frozen rain drops. The
falling graupel and frozen rain drops can strongly enhance the ice generation through ice
multiplication processes (Heymsfield and Willis, 2014) and possibly strengthen the updraft.
Another difference among the three field campaigns is found in the downdrafts. The downdrafts
in HiCu and COPE, which are sampled in mid-latitude convective clouds, are obviously stronger
than those in ICE-T, which was conducted over tropical ocean. This may be because the ambient
relative humidity is low in HiCu and COPE compared to ICE-T, resulting in a faster evaporation
of cloud drops and a stronger cooling effect when ambient air mixes with cloud parcels through
lateral entrainment (Heymsfield et al., 1978). But since the diameters of the downdrafts in ICE-T
are relatively broader (Fig. 4), the air mass fluxes of the downdrafts are not obviously smaller
than that in HiCu and COPE.
Fig. 9 shows the PDFs of the air mass flux for all the drafts sampled at 0–2 km, 2–4 km, 4–6 km
and higher than 6 km. The PDFs are exponentially distributed for the three field campaigns at



different heights, which can be fitted using Eq. (1). The coefficients for the fitted function are
shown in each panel. At 0–2 km, the PDF of the air mass flux in the updrafts is relatively narrow
in ICE-T compared to that in COPE. For the downdraft, the PDF is broader in ICE-T than those
in COPE. As height increases up to 6 km, more updrafts with larger air mass flux are observed in
ICE-T and the PDFs broadens, but in COPE the PDFs remain similar. In HiCu, the PDFs for
updrafts broadens from 2-6 km then remain similar at altitudes higher than 6 km. For downdrafts,
the PDFs are similar at different heights for all the three field campaigns. Among the three field
campaigns, the differences of the PDFs are small for the weak and moderate drafts and are larger
for the strong drafts.

### 421    4.3    Profiles of vertical velocity and air mass flux

Fig. 10 is a Whisker-Box plot showing the profiles of the vertical velocity (a-c) and air mass flux
(d-f) in the drafts based on the three defined thresholds of air mass flux. The solid box includes
all the three different groups of drafts, the dashed boxes excludes the weak drafts, and the dotted
boxes includes strong drafts only. The minimum, 10%, 50%, 90% and the maximum values are
shown in each box. Notice that the vertical velocity and air mass flux in the downdraft is
negative, so the minimum value represents the strongest subsiding parcel, the 10% value
represents the strongest $10^{th}$ percentile subsiding parcel, and the 90% value represents the
weakest $10^{th}$ percentile subsiding parcel. This is opposite to the updraft. In each panel, the
absolute values of the vertical velocities and air mass flux (except the minimum and maximum
ones) are relatively small for the solid boxes.





In Fig. 10a-c, the three definitions of drafts show different intensities in the vertical velocities.
Typically, the 10%, 50% and 90% values in the dotted boxes are 1–2 times larger in magnitude
than those in the solid boxes. However, the profiles of the three definitions of drafts vary
similarly with height for each field campaign. In the updrafts sampled during HiCu (Fig. 10a),
the maximum vertical velocity increases from about 10 m s$^{-1}$ to 18 m s$^{-1}$ with height up to 8 km,
then decreases to 14 m s$^{-1}$ at 8–10 km; the 90% vertical velocity in the solid boxes increases from
4 m s$^{-1}$ to 8 m s$^{-1}$ between 0–10 km. The 10% and 50% vertical velocities in the solid boxes
remain similar between 2–8 km then slightly increase at 8–10 km. The magnitudes of the 10%
and 50% vertical velocities in the solid boxes are about 0.5–0.6 m s$^{-1}$ and 1.8–2.5 m s$^{-1}$. In the
downdrafts, the minimum vertical velocity decreases from –7 m s$^{-1}$ to –12 m s$^{-1}$ up to 8 km and
increases to –9 m s$^{-1}$ at 8–10 km. The 10%, 50 % and 90% values all slightly decrease with
height. In the updrafts sampled during COPE (Fig. 10b), the maximum vertical velocities
increase from 8 m s$^{-1}$ to 23 m s$^{-1}$ between 0–6 km, the 10%, 50% and 90% vertical velocities
increase up to 6 km. The magnitudes are 0.35–0.45 m s$^{-1}$, 1–1.6 m s$^{-1}$, and 2.6–6 m s$^{-1}$ in the solid
boxes, respectively. The minimum vertical velocity in the downdrafts intensifies from –5 to –10
m s$^{-1}$ with height up to 4 km, then remains similar at 4–6 km. The strongest updraft and
downdraft are observed at 4–6 km, about 23 m s$^{-1}$ and -10 m s$^{-1}$, respectively. In the updrafts
sampled during ICE-T (Fig. 10c), the maximum vertical velocities increase with height from 5.5
m s$^{-1}$ to 25 m s$^{-1}$ up to 6 km, then slightly decrease at 6–8 km. The 90% value increases from 2 to
6 m s$^{-1}$ between 0-4 km, then remains similar at higher levels. The 10% and 50% values, which
are about 0.32–0.6 m s$^{-1}$ and 0.8–1.8 m s$^{-1}$ in the solid boxes, respectively, do not show an
obvious trend with height. In the downdrafts the minimum vertical velocity increases from –6 m
s$^{-1}$  to –5 m s$^{-1}$ between 0 km and 4 km, and decreases from –5 m s$^{-1}$  to –18 m s$^{-1}$ between 4 km



and 8 km. The 10%, 50% and 90% values tend to decrease or remain similar at first and then
increase with height. The peak (~25 m s$^{-1}$) and the minimum (~-18 m s$^{-1}$) vertical velocities are
observed at 4–6 km and 6–8 km, respectively.
To summarize, vertical velocity in the drafts varies differently with height in the three field
campaigns. Generally, the maximum and 90% vertical velocities in the updrafts are greater in
COPE or ICE-T than in HiCu, while the median vertical velocities are the greatest in HiCu and
weakest in ICE-T. Stronger downdrafts are often observed in HiCu and COPE compared to those
in ICE-T. The weak, moderate and strong drafts have similar variations of the vertical velocity
with height, but the magnitudes are the smallest when including all the drafts and become larger
if the weak drafts are excluded. The 10%, 50% and 90% vertical velocities in updrafts and
downdrafts over tropical ocean (ICE-T) observed in this study generally have similar magnitudes
to those shown in previous studies (e.g. LeMone and Zipser, 1980; Lucus and Zipser, 1994). But
strong updrafts (downdrafts) in excess of 20 m s$^{-1}$ (–10 m s$^{-1}$) are also observed in this study,
which are not shown in pervious aircraft observations. This finding is consistent with recent
remote sensing observations (e.g. Heymsfield et al., 2009). The updrafts and downdrafts in
convective clouds over land shown in this study (HiCu) are weaker than those shown by Byers
and Braham (1949) and Heymsfield et al. (2009), possibly because HiCu was conducted over the
High Plains.
Fig. 10d-f shows the profiles the air mass flux statistics for the drafts sampled during the three
field campaigns. As expected, the absolute values of the air mass flux are relatively small if all
the drafts are included (dotted boxes), and become larger if the drafts with small air mass flux are
excluded. However, the variations of the air mass flux with height are similar for the three





different definitions in each panel. As determined by the three thresholds, the minimum absolute
values in the solid boxes are about 10 times smaller than those in the dashed boxes and about 50
times smaller than those in the dotted boxed; for the 10%, 50%, 90% and the maximum absolute
values, the differences among the three type of boxes become smaller. In HiCu, the air mass flux
does not show an obvious trend with height. In the updraft, the 10%, 50% and 90% values
remain similar at different height ranges. The maximum air mass flux increases from 2–6 km,
then decreases with height. The peak value is about $1.3 \times 10^4$ kg m$^{-1}$ s$^{-1}$, found at 4–6 km. The air
mass flux in the downdrafts has relatively larger variability, especially for the minimum values.
The strongest downdraft in terms of air mass flux (about $-1.2 \times 10^4$ kg m$^{-1}$ s$^{-1}$) is found at 4–6 km,
but this is probably due to a specific case since the 50% and 90% values are similar to those at
the other height ranges. In COPE, the 90% and the maximum air mass flux in the updraft tend to
increase with height, while the 10% and 50% values are similar at different height ranges. For
the downdraft, the minimum air mass flux decreases between 0–2 km and remains similar at 4-6
km. The 10%, 50% and 90% values are similar at different height ranges. The strongest updrafts
and downdrafts in terms of air mass flux are observed at 4–6 km and 2–4 km, about $1.8 \times 10^4$ kg
m$^{-1}$ s$^{-1}$ and $-2.8 \times 10^3$ kg m$^{-1}$ s$^{-1}$. In ICE-T, the maximum air mass flux in the updraft increases
with height up to 6 km, then decreases at 6–8 km. The 10%, 50% and 90% values in the updraft
and downdraft intensify from 0-4 km and decrease or remain similar at higher levels. The
strongest updraft ($3 \times 10^4$ kg m$^{-1}$ s$^{-1}$) and downdraft ($-3.5 \times 10^3$ kg m$^{-1}$ s$^{-1}$) are observed at 4–6 km
and 0–2 km, respectively. The minimum value is probably due to a specific case because the
10%, 50% and 90% values at 0–2 km are larger or similar to those at the other heights.
To summarize, the air mass flux varies with height differently for the three field campaigns. For
updraft, the maximum air mass flux is of the order of $10^4$ kg m$^{-1}$ s$^{-1}$, and the median values for





the three different types of boxes are typically ~100 kg m$^{-1}$ s$^{-1}$, ~200 kg m$^{-1}$ s$^{-1}$ and ~1000 kg m$^{-1}$
s$^{-1}$, respectively. The air mass flux in the downdrafts is a few times smaller in magnitude than
those in the updrafts, but extreme strong downdraft on the order of $10^4$ kg m$^{-1}$ s$^{-1}$ may be
observed in some specific cases. Compared to previous studies, the air mass flux in this study
shows similar magnitudes, but the vertical dependences are different. Lucas and Zipser (1994)
show that the convection off tropical Australia intensifies with height from 0 to 3 km, then
weakens with height in terms of air mass flux. Anderson et al. (2005) shows that updrafts and
downdrafts over the tropical Pacific Ocean intensify with height up to 4 km, then weaken at
higher levels in terms of air mass flux. In the present study, the strongest updrafts and
downdrafts are observed at higher levels for all the three field campaigns.

**4.4      Composite structure of vertical velocity**
Fig. 11 shows the composite structure of the vertical velocity as a function of the normalized
diameter for the updrafts and downdrafts with air mass flux $\geq$ 10 kg m$^{-1}$ s$^{-1}$, 100 kg m$^{-1}$ s$^{-1}$ and
500 kg m$^{-1}$ s$^{-1}$ in magnitude. As expected, the draft as a whole is weaker if all the drafts are
included in the calculation and becomes stronger if the drafts with small air mass flux are
excluded. In HiCu, when all weak, moderate and strong updrafts are included (red curves), the
vertical velocity near the center is about 1.7 m s$^{-1}$. When only moderate and strong updrafts are
included (green curves), the vertical velocity near the center is ~2.4 m s$^{-1}$. When all the updrafts
with air mass flux smaller than 500 kg m$^{-1}$ s$^{-1}$ in magnitude are excluded, the absolute values of
the vertical velocity near the center increase to ~3.4 m s$^{-1}$. The vertical velocity in downdrafts is
about 0.2 m s$^{-1}$ smaller in magnitude than that in updrafts. The structures of the vertical velocity



in COPE are quite similar to those in HiCu, in both shape and magnitude, especially for the red
and green curves. The blue curves have relatively larger variations due to the small sample size.
These variations reveal the complicated structure in some drafts. In ICE-T, the shapes of the
vertical velocity structures are similar to those in HiCu and COPE, but the magnitudes are
smaller, which suggests that statistically more weak drafts are found in ICE-T, although the peak
vertical velocity is observed in ICE-T. This is consistent with Fig. 10. In Fig. 11, if the
magnitude of the vertical velocity is normalized, the structures of the three defined classes of
updraft and downdraft among the three field campaigns will be very similar.
In this composite analysis based on in-situ measurements, the penetration direction has no
obvious impact on the vertical velocity structure, whether the aircraft penetrates along or across
the horizontal wind. For convective cloud, wind shear has a large impact on the cloud evolution
(Weisman and Klemp 1982); however, aircraft data are insufficient to reveal the wind shear
impact, because each penetration is made at a single level and the aircraft does not always
penetrate through the center of the draft. Remote sensing data can be helpful to study the two-
dimensional or three-dimensional structures of the vertical velocity in convective clouds (e.g.
Wang and Geerts, 2013). Thus, in-situ measurements as well as remote sensing measurements
are needed to further analyze the wind shear impact.

**4.5      Vertical air motion characteristics as clouds evolve**
Fig. 12 shows the profiles of the vertical velocity (a-c) and the air mass flux (d-f) for the updraft
and downdraft in the convective clouds with different cloud top heights (CTH). Here, all weak,





moderate and strong updrafts are included. Different colors represent the clouds with different
CTHs. These profiles can generally reveal the change of vertical velocity and air mass flux as the
clouds evolve. The key point presented in Fig. 12 is that the peak vertical velocity and air mass
flux is observed at higher levels as the clouds evolve. For clouds with CTHs lower than 4 km
(red boxes), the maximum vertical velocity is observed at 2–4 km. When the cloud become
deeper, the vertical velocity and air mass flux are stronger at higher levels. This is to be expected,
because all the data analyzed in this paper are collected from isolated convective clouds, so the
convective bubbles keep ascending as the clouds evolve. MCSs may have different
characteristics of vertical air motion because there is continuous low level convective source.
The maximum vertical velocity is observed within 2 km below cloud top; this is consistent with
Doppler velocity images measured by WCR (e.g. Fig. 2b), which show the typical strongest
updraft is observed 1–1.5 km below cloud top. The strongest downdrafts are sometimes observed
more than 2 km below cloud top. The 10%, 50% and 90% values do not have obvious trend as
the clouds evolve, especially in HiCu and ICE-T, possibly because of the increasing contribution
from moderate and weak drafts as the clouds become deeper and broader (Fig. 6 and 7).
Generally, in HiCu and ICE-T the drafts intensify as the clouds evolve, but this is not found in
COPE, maybe because most of the penetrations were made near the cloud top, rather than in the
strongest portion of a draft. Since the vertical resolution of aircraft in-situ data is poor, more data,
including remote sensing measurements, are needed to better understand the evolution of the
vertical velocity in convective clouds as they go through the different stages..

**5.    Conclusions**





The vertical velocity and air mass flux in convective clouds are statistically analyzed in this
study using aircraft data collected from three field campaigns, HiCu, COPE and ICE-T,
conducted over mid-latitude High Plains, mid-latitude coastal area and tropical ocean. Three
thresholds of air mass flux are selected to delineate draft: 10 kg m$^{-1}$ s$^{-1}$, 100 kg m$^{-1}$ s$^{-1}$ and 500 kg
m$^{-1}$ s$^{-1}$ in magnitude. The main findings are as follows.
1)    Small-scale updrafts and downdrafts in convective clouds are often observed in the three
field campaigns. More than 85%, 90% and 74% of the updrafts are narrower than 500 m in HiCu,
COPE and ICE-T, respectively, and more than 90 % of the downdrafts are narrower than 500 m
in the three field campaigns combined. These small scale drafts make significant contributions to
the total air mass flux. Updrafts narrower than 500 m contribute 20%−35% of the total upward
flux, and downdrafts narrower than 500 m contribute 50%−65% of the total downward air mass
flux.
2)    In terms of the air mass flux, the weak and moderate drafts make an important
contribution to the total air mas flux exchange. Generally, the number of drafts increases with
cloud diameter. For many narrow clouds, the weak and moderate drafts dominate and contribute
most of the total air mass flux. For broader clouds, the stronger updrafts contribute most of the
total air mass flux, but the contribution from weak and moderate drafts increases as the cloud
evolves.
3)    PDFs and profiles of the vertical velocity are provided for the three defined types of
drafts. In all the height ranges, the PDFs are roughly exponentially distributed. At the lowest
level, the PDFs of the vertical velocity are relatively narrow, and broaden with height. For the
updrafts, the PDFs of the vertical velocity are broader in ICE-T and COPE, while for the



downdrafts the PDFs of the vertical velocity are broader in HiCu and COPE. The profiles show
that updrafts are stronger in ICE-T and COPE than in HiCu, and downdrafts are stronger in HiCu
and COPE compared to ICE-T.
4)    PDFs and profiles of the air mass flux are provided for the drafts. The PDFs are similarly
exponentially distributed at different heights. For updrafts, the PDFs are broader in ICE-T than in
HiCu and COPE, but for downdrafts the PDFs are broader in HiCu and COPE than in ICE-T. In
the updrafts, the maximum air mass flux has an order of $10^4$ kg m$^{-1}$ s$^{-1}$. The air mass flux in the
downdrafts are typically a few times smaller in magnitude than those in the updrafts.
5)    The composite structures of the vertical velocity in the updrafts and downdrafts have
similar shapes for the three field campaigns: the vertical velocity is the strongest near the center,
and weakens towards the edges. On average, the updrafts have similar intensity across the three
field campaigns, while for downdrafts the vertical velocity is the weakest in ICE-T and stronger
in HiCu and COPE.
6)    The change of vertical air motion characteristics as the cloud evolves are briefly
discussed. Generally, the strongest portion of a draft ascends with height as the cloud evolves.
The maximum vertical velocity is observed within 2 km below cloud top; the downdrafts are
sometimes stronger at levels more than 2 km below cloud top.
Based on the aircraft observations from three field campaigns, this study provides quantitative
analyses of the vertical air motion characteristics in isolated convective clouds, compares the
differences of vertical velocity and air mass flux among the different field campaigns, and shows
the importance of small-scale updrafts and downdrafts. The results are useful to evaluate model



simulations and improve parameterizations in models. To better understand the differences of the
vertical air motions among different convective clouds and the evolution of the updrafts and
downdrafts in convective clouds more data are needed.

**Acknowledgments**
This work is supported by National Science Foundation Award: AGS-1230203 and AGS-
1034858, the National Basic Research Program of China under grant no. 2013CB955802 and
DOE Grant DE-SC0006974 as part of the ASR program. The authors acknowledge the crew of
NCAR C-130 and University of Wyoming King Air for collecting the data and for providing
high-quality products.



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



Table 1. Number of penetrations, time in clouds and flight length in clouds sampled at 0–2 km, 2–4 km, 4–6 km, 6–8 km and 8–10 km MSL in HiCu, COPE and ICE-T.

| Height (km MSL) | HiCu | | | COPE | | | ICE-T | | |
|---|---|---|---|---|---|---|---|---|---|
| | Number of penetrations | Time in clouds (min) | Length in clouds (km) | Number of penetrations | Time in clouds (min) | Length in clouds (km) | Number of penetrations | Time in clouds (min) | Length in clouds (km) |
| 8–10 | 43 | 12 | 79 | | | | | | |
| 6–8 | 565 | 122 | 789 | | | | 132 | 52 | 423 |
| 4–6 | 596 | 104 | 653 | 207 | 39 | 244 | 299 | 116 | 895 |
| 2–4 | 373 | 50 | 274 | 378 | 86 | 486 | 34 | 10 | 73 |
| 0–2 | | | | 219 | 40 | 211 | 197 | 27 | 167 |





Table 2. Number of updrafts and downdrafts sampled at 0-2 km, 2-4 km, 4-6 km, 6-8 km and 8-10 km in HiCu, COPE and ICE-T.

Three numbers are given for the updraft and downdraft at each level, respectively, according to the three different definitions: weak, moderate and strong.

| Height (km) | | HiCu | | COPE | | ICE-T | |
|---|---|---|---|---|---|---|---|
| | | Updraft | Downdraft | Updraft | Downdraft | Updraft | Downdraft |
| 8-10 | weak | 66 | 100 | | | | |
| | moderate | 52 | 44 | | | | |
| | strong | 44 | 17 | | | | |
| 6-8 | weak | 818 | 763 | | | 382 | 372 |
| | moderate | 559 | 540 | | | 175 | 136 |
| | strong | 287 | 130 | | | 102 | 23 |
| 4-6 | weak | 748 | 668 | 290 | 184 | 858 | 671 |
| | moderate | 522 | 389 | 232 | 193 | 425 | 329 |
| | strong | 343 | 48 | 135 | 51 | 266 | 73 |
| 2-4 | weak | 311 | 235 | 568 | 424 | 49 | 47 |
| | moderate | 271 | 84 | 467 | 434 | 51 | 51 |
| | strong | 149 | 7 | 188 | 101 | 32 | 10 |
| 0-2 | weak | | | 368 | 192 | 319 | 205 |
| | moderate | | | 266 | 90 | 234 | 104 |
| | strong | | | 96 | 9 | 60 | 7 |





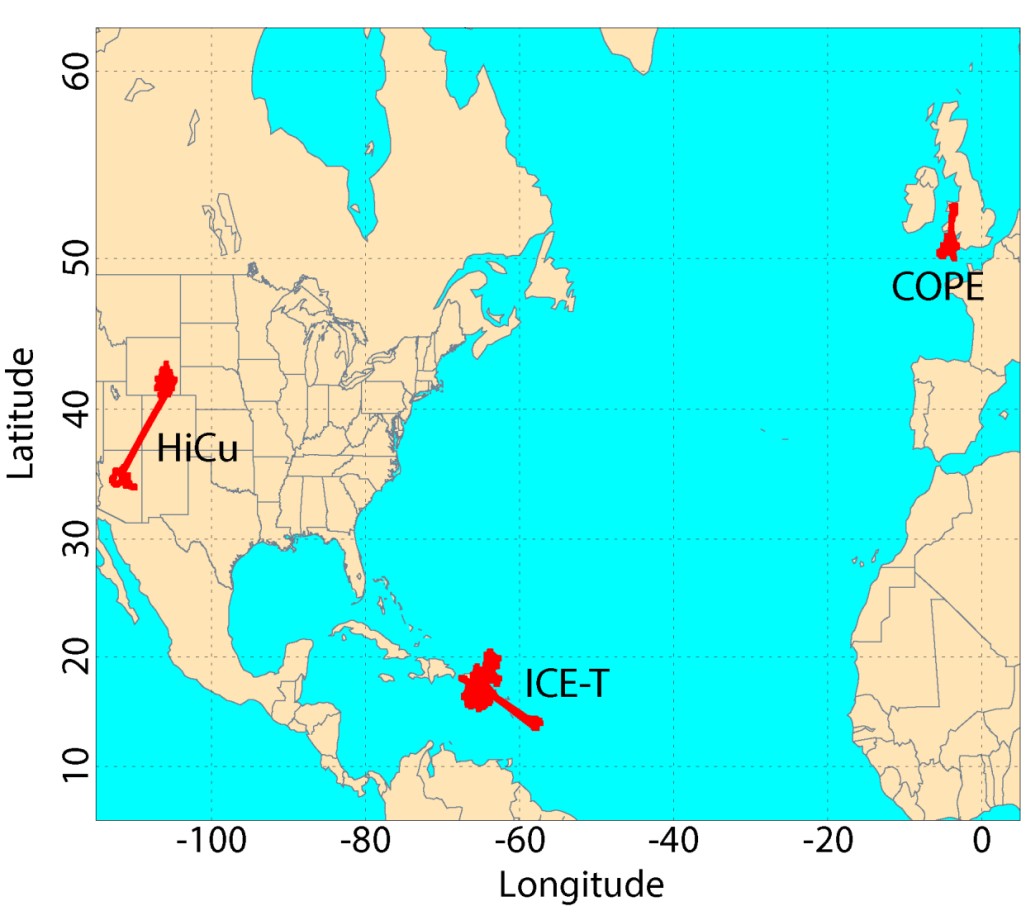

Figure 1. Flight tracks for the three field campaigns: HiCu, COPE and ICE-T.



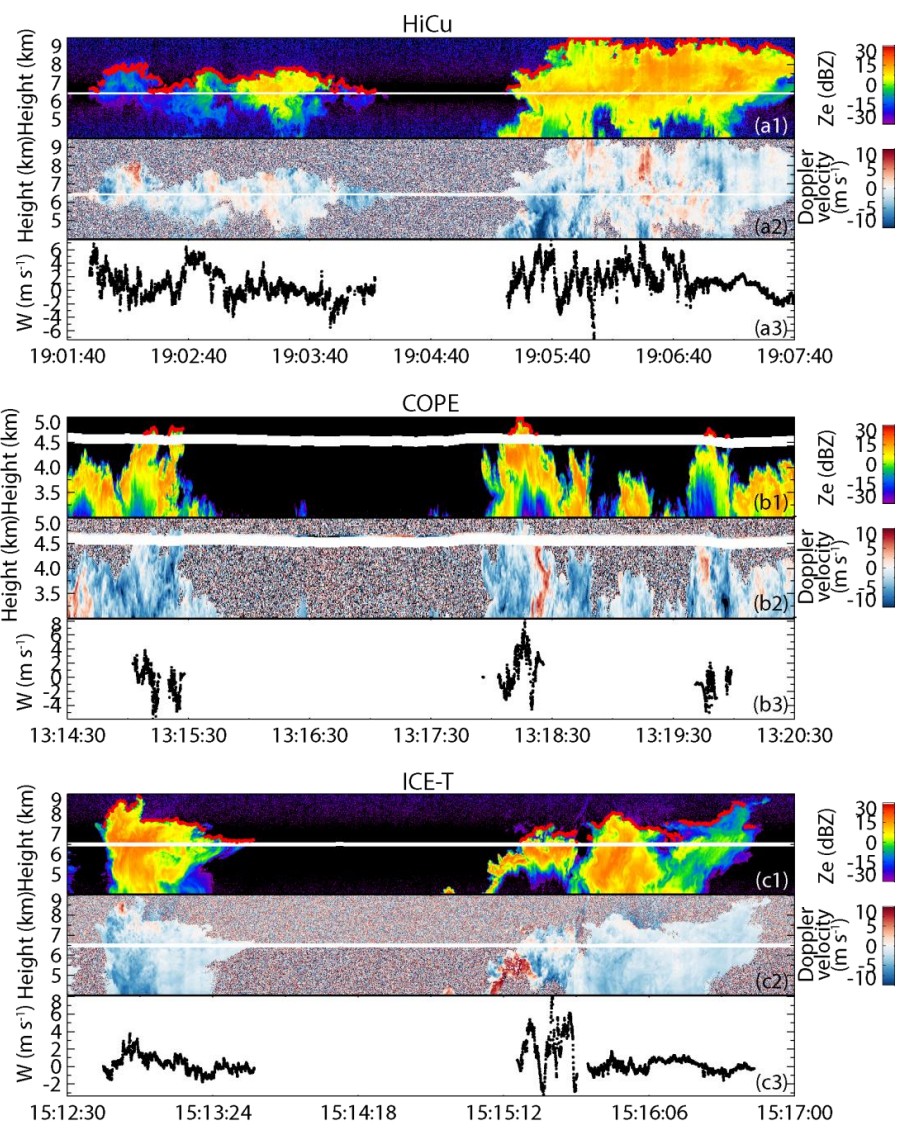

Figure 2. Examples of radar reflectivity, Doppler velocity and 25-Hz in-situ vertical velocity measurements for the convective clouds sampled in HiCu, COPE and ICE-T. The red dots in (a1), (b1) and (c1) are the cloud tops estimated by WCR.




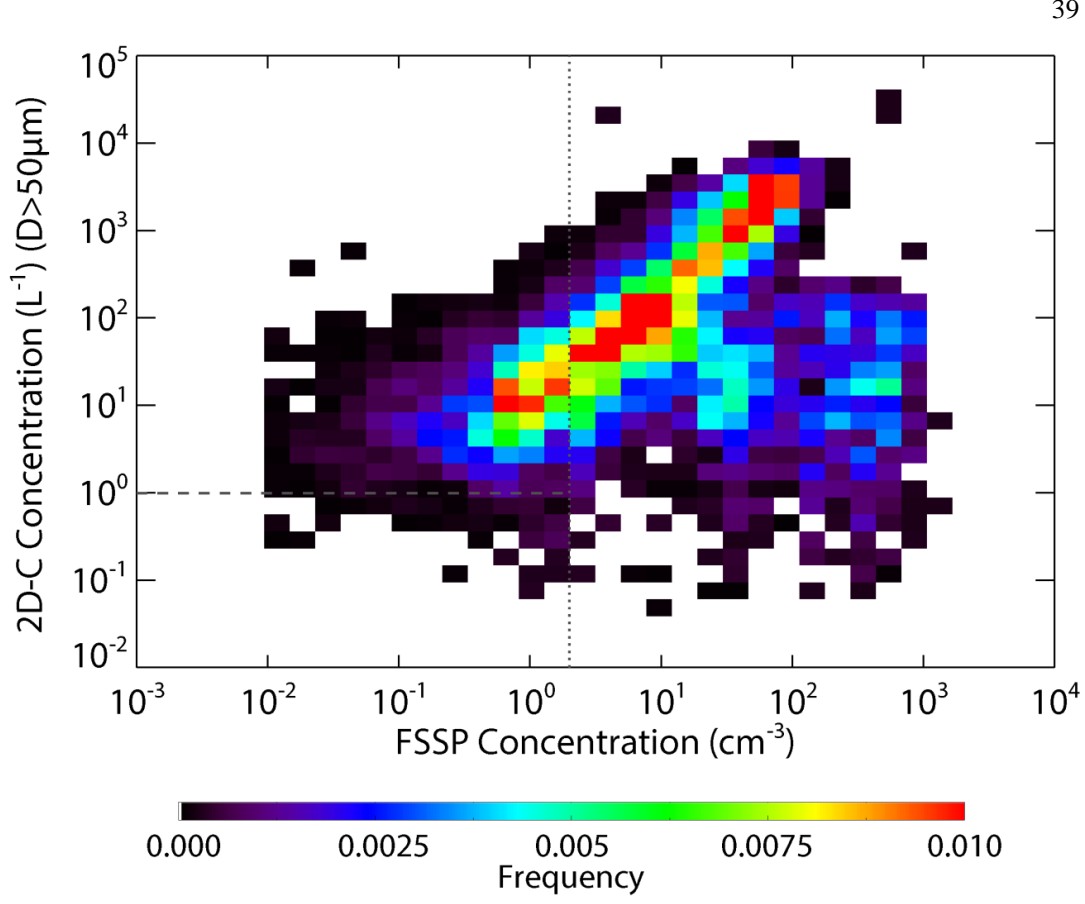

Figure 3. Occurrence distributions as a function of the particle concentrations measured by FSSP versus the concentrations of the particles ≥ 50 μm in diameter measured by 2D-C in the clouds identified by WCR reflectivity. The dashed and dotted lines indicate the FSSP concentration equal 2 cm$^{-3}$ and the 2D-C concentration equal 1 L$^{-1}$, respectively.







Figure 4. PDFs of the diameters for the updrafts and downdrafts sampled at 0–2 km, 2–4 km, 4–6

km and higher than 6 km. The numbers shown in each panel are the coefficients of the fitted

exponential function (Eq. 1).



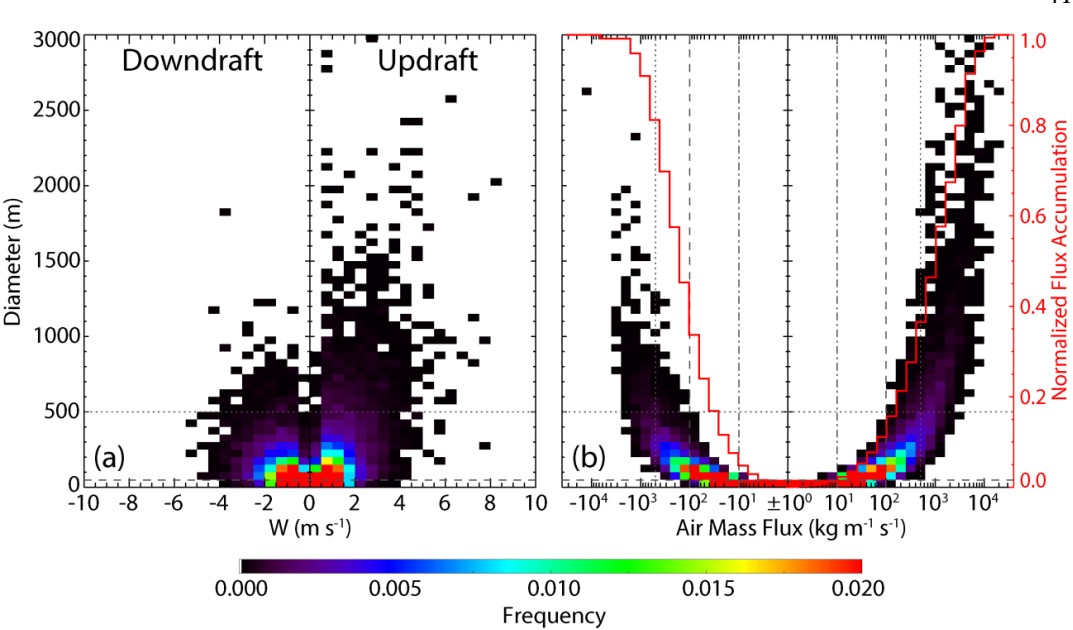

Figure 5. Occurrence distributions as (a) a function of diameter and mean vertical velocity, and (b) a function of diameter and air mass flux for all updrafts and downdrafts. The normalized accumulation flux is also shown by the red curves. The horizontal dotted and dashed lines in (a) and (b) indicate the draft diameter equal 500 m and 50 m, which are used as the diameter thresholds to identify a "draft" in previous studies and in this study, respectively. The vertical dash-dotted, dashed and dotted lines in (b) indicate air mass flux equal 10 kg m$^{-1}$ s$^{-1}$, 100 kg m$^{-1}$ s$^{-1}$ and 500 kg m$^{-1}$ s$^{-1}$ in magnitude, respectively, which are the thresholds used to delineate the three different groups of draft.





Figure 6. (a) Average number and (b) occurrence frequency of updrafts as a function of air mass

flux observed in penetrations with length < 1 km (solid), 1-10 km (dashed) and >10 km (dotted).

The result is a composite of HiCu, COPE and ICE-T.





Figure 7. Average percentile contribution to total upward air mass flux by the weak (red),

moderate (green) and strong (blue) updrafts delineated in this study. The result is a composite of

HiCu, COPE and ICE-T.





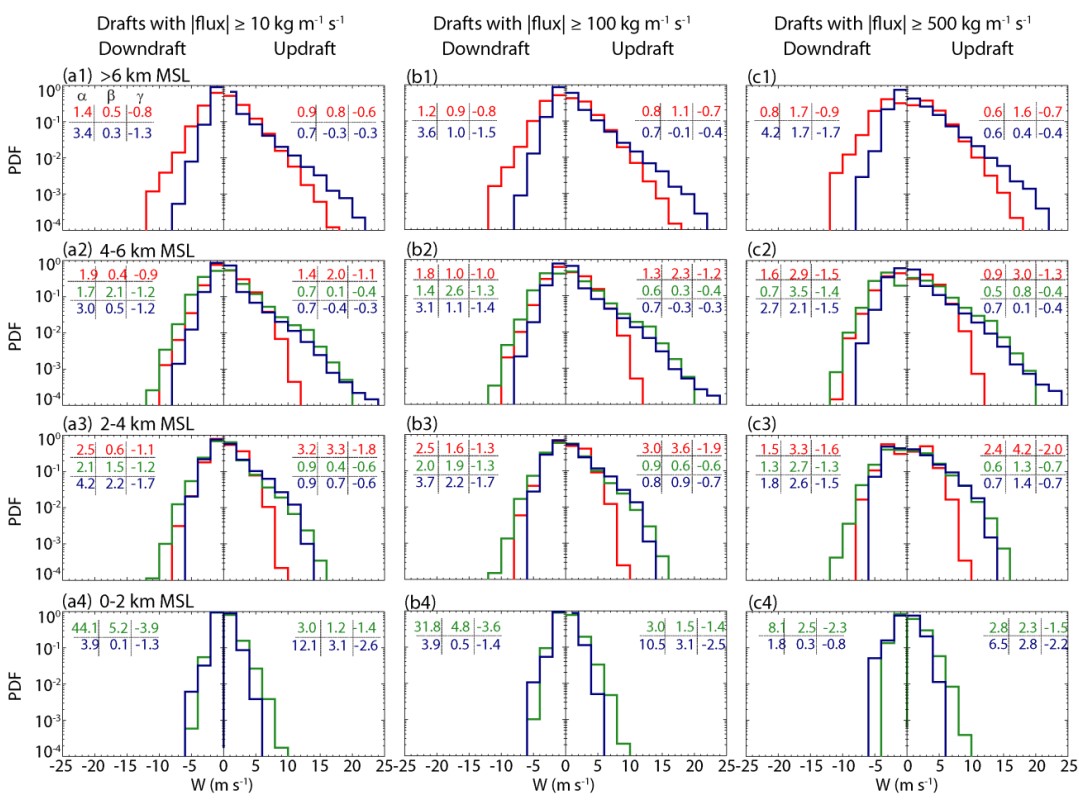

Figure 8. PDFs of the 25-Hz vertical velocity for the updrafts and downdrafts with air mass flux $\geq$ (a) 10 kg m$^{-1}$ s$^{-1}$, (b) 100 kg m$^{-1}$ s$^{-1}$ and (c) 500 kg m$^{-1}$ s$^{-1}$ in magnitude, sampled at 0–2 km, 2–4 km, 4–6 km and higher than 6 km. The numbers shown in each panel are the coefficients of the fitted exponential function (Eq. 1).



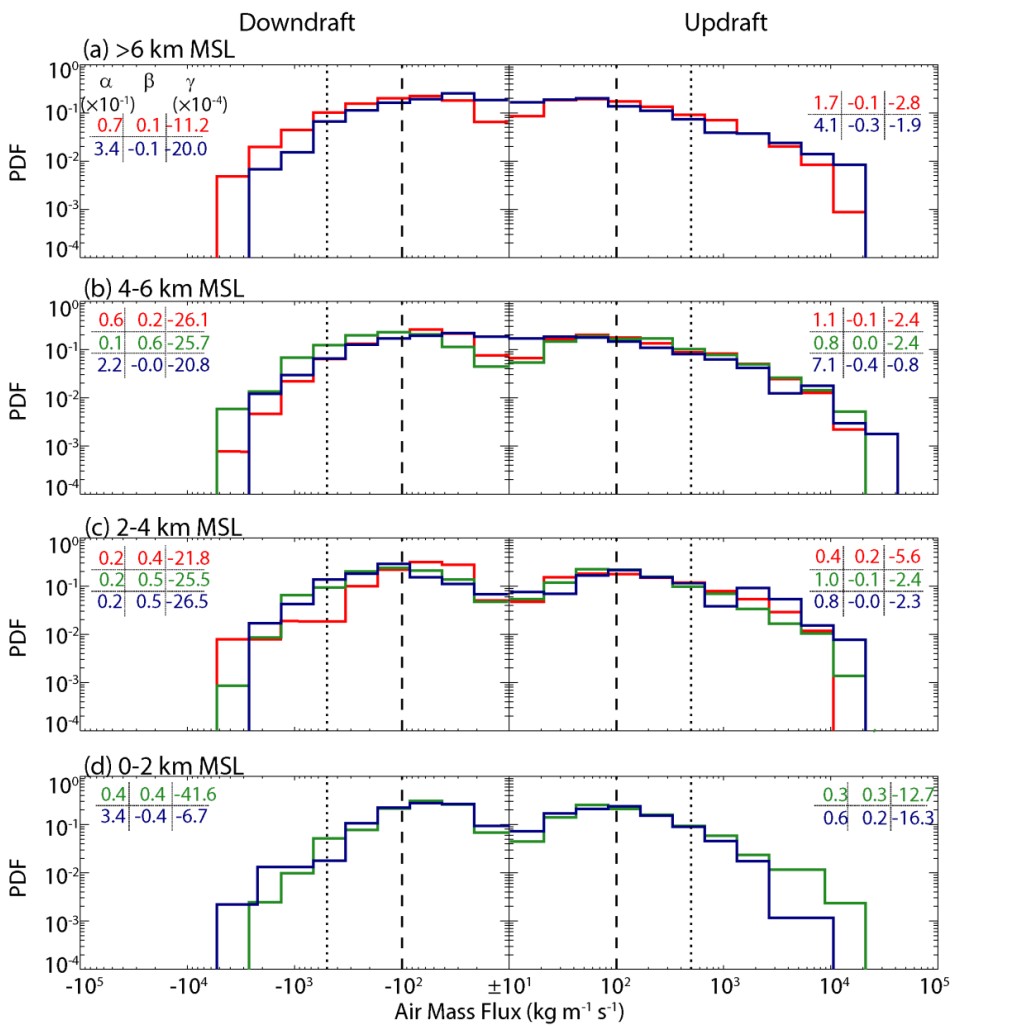

Figure 9. PDFs of the air mass flux for the updrafts and downdrafts sampled at 0–2 km, 2–4 km, 4–6 km and higher than 6 km. The three thresholds of the air mass flux ($\pm 10$ kg m$^{-1}$ s$^{-1}$, $\pm 100$ kg m$^{-1}$ s$^{-1}$ and $\pm 500$ kg m$^{-1}$ s$^{-1}$) are shown by the solid (overlaps with the central y-axis in each panel), dashed and dotted lines. The numbers shown in each panel are the coefficients of the fitted exponential function (Eq. 1).







Figure 10. Profiles of (a-c) the vertical velocity and (d-f) air mass flux for all the updrafts and downdrafts sampled at 0–2 km, 2–4 km, 4–6 km, 6–8 km and 8–10 km. The dotted, dashed and solid boxes represent for the drafts with air mass flux ≥ 10 kg m$^{-1}$ s$^{-1}$, 100 kg m$^{-1}$ s$^{-1}$ and 500 kg m$^{-1}$ s$^{-1}$ in magnitude, respectively.




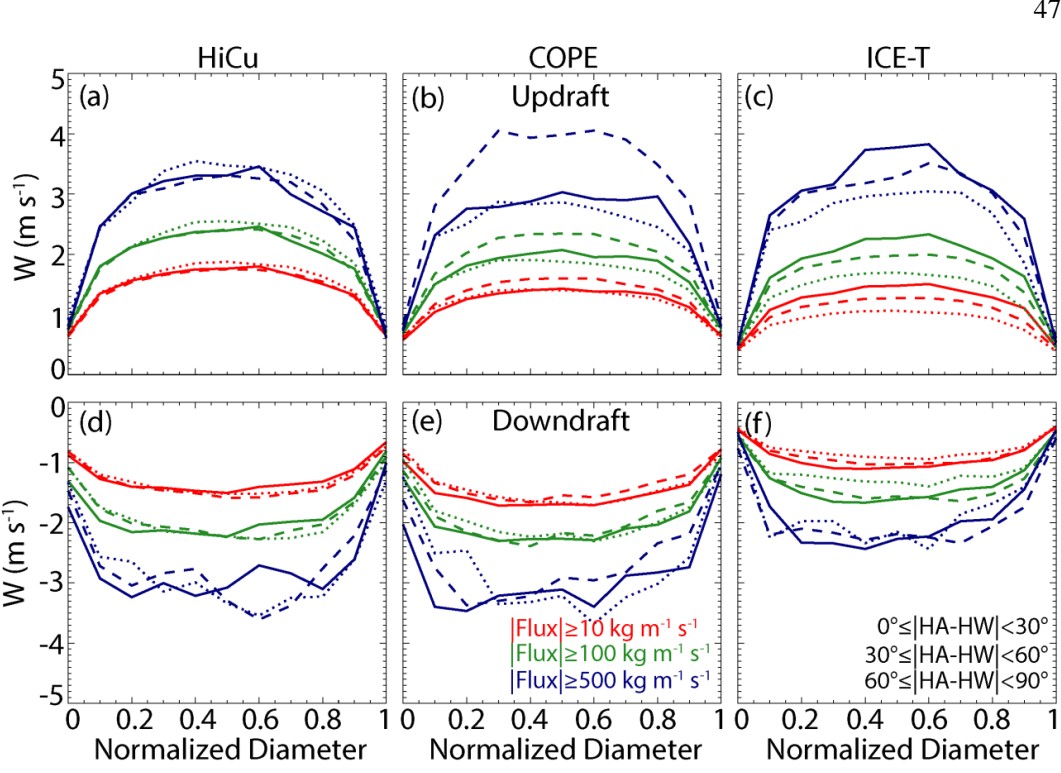

Figure 11. Composite structure of the vertical velocity as a function of the normalized diameter for the updrafts and downdrafts with air mass flux ≥ 10 kg m$^{-1}$ s$^{-1}$, 100 kg m$^{-1}$ s$^{-1}$ and 500 kg m$^{-1}$ s$^{-1}$ in magnitude. The solid, dashed and dotted curves represent penetrations with the heading angles (HA) 0°–30°, 30°–60° and 60°–90° from the horizontal wind (HW) directions, respectively. The 0 and 1 coordinates on the x-axis indicate the upwind and downwind sides of the draft.





Figure 12. Profiles of (a-c) the vertical velocity and (d-f) the air mass flux for the updraft and downdraft with air mass flux ≥ 10 kg m$^{-1}$ s$^{-1}$ in magnitude. The red, orange, green and blue boxes represent clouds with cloud top heights of 0-4 km, 4-6 km, 6-8 km and higher than 8 km.