# Peer review of "Characteristics of Vertical Air Motion in Isolated Convective Clouds"

_Atmospheric Chemistry and Physics, 2015_

## Referee Comment (RC1) · Anonymous Referee #1 · 4 Mar 2016

This paper provides a valuable contribution since there is not a huge amount of updraft information of this type in the literature. The results are valuable for a variety of reasons such as for providing better realism for numerical models on convection vertical motion scales and magnitudes. The results are also very useful for microphysical studies such as rain/snow growth mechanisms that require a vertical motions as a key input. The authors provide a good summary of past work and the manuscript in general is well written except for numerous typographical errors and some poorly worded sentences. The technical details are sufficient for the material presented. There is too much detail in some sections, and only the key points should be included (e.g., section 4.3).

The paper deals with what shallow to moderate convection. The authors need to add some discussion in the abstract and conclusions on the fact that the measurements presented are still a biased sample of convection. Are the measurements truly representative of all convection in the three regions presented, or did for example the planes used stay away from stronger, and/or deeper convection, or ones with higher reflectivities. What the paper points out is that there are some similarities between the regions, but that there is really a wide variety of convective types over the globe. This is a good point to make in the paper that there are few measurements of this sort so they are greatly needed, but they represent specific regions and types of convection, more from other regions are needed, and one should not interpret the results that these results can be generalized globally. A few summary sentences (abstract, intro, and conclusions) on this point would make the paper better.

While I find the paper quite interesting, there could be more connection between the convection dynamics and microphysics. Processes such as mixing are barely mentioned in the text. It would be interesting for example to make connections between the updraft characteristics such as mass fluxes, diameters, and entrainment.

Technical and other details:

Lines 128-138: What are typical reflectivities in the convection. I know this will be from the W-band radar but this information would still be useful.

Lines 171: Define strong updrafts since these are still relatively weak compared to deeper convection.

Line 189: This accuracy (0.2 m/s) is quite good. Is there any chance there are biases rather than random errors on the vertical velocity?

Line 209: Footnote. This is roughly the sensitivity of CloudSat. Is this the reason -30 dBZ was chosen since there will be cloud at lower reflectivities?

Lines 233-235: How do you know the 2D symmetry of the updraft since you might not fly through the peak up and downdrafts? Both the W-band radar and in situ measurements will not tell you this.

Line 236: "there is no" should be "there are no"

Line 238: Excluding MCS biases the results. This comes down to emphasis in this paper on small to moderate convection rather than deep convection in MCSs. Might mention this to keep the scope of your study in perspective.

Figure 4: need labels for field experiment associated with each color.

Lines 264-268: It might be useful to plot one example of a trace through one of the updraft/downdraft penetrations. This would be helpful to understand some of the averaging performed.

Line 268: "turbulences" to "turbulence"

Line 300: "convections" to "convection"

Line 294: "strong draft" – I would again put this in perspective since it is strong in your study, but not necessarily strong with respect to MCS updrafts for example.

Line 323-324: Again, this point should be more prominent in the paper.

Lines 386-387: This statement should be in the summary/conclusions since these measurements are important but we need a lot more.

Line 397: "results" to "result"

Line 403: "relatively" to "relative"

Figure 10 and similar plots: I find these plots a little complicated and confusing but probably acceptable for publication. I can't think of an alternative but possibly there is a better way to plot.

Line 422: Some of the statements in this section should go in a summary and conclusions section. Points like lines 468-473 are important summary statements. You should consider pulling some of the summary points like this and putting them in the conclusions.

Line 469: "pervious" to "previous"
Lines 509-510: "While in this study" should be something like "In contrast, this study shows the strongest...."

Line 522: "When exclude" to "When we exclude"

Line 536: "convective cloud" to "convective clouds"

Lines 539-540: Can you say anything about the two-dimensionality of drafts from the remote sensing data?

Line 553: "with expectation" to "expected"

Line 564: "Since ....to better". This is an obvious fact. May want to just say that the aircraft just provides a line of data through drafts, and not vertical information unless the plane makes multiple passes through the same cell.

Line 568: in the Summary section, you should reiterate the criteria for considering up/downdrafts, i.e., >xx m/s.

Line 596: Flux calculations assume two-dimensionality of drafts and this might not be the case. Should mention this as a weakness in the study, i.e., using a single line penetration through drafts to make flux calculations.

---

## Referee Comment (RC2) · S. Collis (Referee) · 10 Mar 2016

I always like to see an in-depth study of vertical motions in the atmosphere because, as the authors point out, understanding these is vital to improving our understanding of (and hence modeling capabilities) many processes influenced by vertical motions.

First, before I get to the science, this document was not ready for submission in any form. It is riddled with typographical errors making it very difficult to get to the science. I started to list them but, frankly, this is the job of an editorial service, something I recommend the author take advantage of. For example: "The COPE project was conducted from 03 July to 21 August, 2013". This is not English.. "The COPE project was conducted from the 3rd of July to the 21st August, 2013".. Write in English not in code.

I have two broad areas of concern with this manuscript:

[Figure]

1) The authors do not address the idea of sample size or sample bias OR more importantly geometric issues of sampling, in a line, a 2/3D object (being an updraft core). See Giangrande et al 2013 for a discussion of issues with profiler systems and angle of attack.. Basically if you dissect an updraft core how do you know if you hit the strongest part of the updraft? Furthermore, up until the end, the idea of selection bias is not addressed. Even the C-130 will avoid the strongest cores. You can not build a PDF out to the tail from aircraft measurements..

You can, as the paper did somewhat, look at intrinsic updraft properties. But you can not look at the distribution. I am somewhat disappointed , given the brief reference to microphysical measurements, that the authors did not relate vertical motions to microphysical properties of the updraft cores. This is something in-situ platforms are uniquely capable of doing. Also, in the literature review of methodologies for measuring vertical motions the authors neglect scanning radar measurements such as those shown in Collis et al 2013 and Nicol et al 2015 (not to mention a raft of airborne radar measurements from the NOAA p3 (look for papers from Jorgensen) and other aircraft that use the vertical plus 45 degree tilt methods..

2) This comment relates to a specific question asked by the Journal in its review criteria "Are substantial conclusions reached?". I am deeply concerned by the authors attempt to relate the three field programs and say something about maritime versus continental convection. For one, the author did not put the cases into context.. What was the CAPE for various cases? etc.. A selection of clouds at each campaign a climatology does not make. While the author caveats his comparison even the attempt to contrast the different regime is dangerous. For one, as mentioned, the strongest cores in the region of HiCu would all but destroy even the C-130 (See the various photos associated with the Byers et al study of hail damage). To attempt to make a comparison, then state it goes contrary to common conception (Continental » Maritime) and then turn around and say "we did not sample the strongest updrafts in the continental case" is disingenuous.

So negatives out of the way, one of the things that redeem the paper is the the focus on updraft shape and how that varies with height. Personally I find this very interesting as not only does the mass flux of a plume influence transport but the vertical velocity within determines many microphysical aspects. ie a plume that starts thin and then expand for the same mass flux would have lower vertical velocities aloft influencing processes like Hallett-Mossop splintering etc.. (and associated latent feedbacks).. The paper should focus more on this and the *intrinsic* differences. Things that are co-varying and less susceptible to sampling and decision bias.

Giangrande, S. E., S. Collis, J. Straka, A. Protat, C. Williams, and S. Krueger, 2013: A Summary of Convective Core Vertical Velocity Properties Using ARM UHF Wind Profilers in Oklahoma. J. Appl. Meteor. Climatol., doi:10.1175/JAMC-D-12-0185.1. http://dx.doi.org/10.1175/JAMC-D-12-0185.1 (Accessed July 16, 2013).

Nicol, J. C., R. J. Hogan, T. H. M. Stein, K. E. Hanley, P. A. Clark, C. E. Halliwell, H. W. Lean, and R. S. Plant, 2015: Convective updraught evaluation in high-resolution NWP simulations using single-Doppler radar measurements. Q.J.R. Meteorol. Soc., 141, 3177–3189, doi:10.1002/qj.2602.

Collis, S., A. Protat, P. T. May, and C. Williams, 2013: Statistics of Storm Updraft Velocities from TWP-ICE Including Verification with Profiling Measurements. J. Appl. Meteor. Climatol., 52, 1909–1922, doi:10.1175/JAMC-D-12-0230.1.

---

## Author Comment (AC1) · 23 May 2016

Reviewer's comments in black, replies in blue.

Comments from reviewer

This paper provides a valuable contribution since there is not a huge amount of updraft information of this type in the literature. The results are valuable for a variety of reasons such as for providing better realism for numerical models on convection vertical motion scales and magnitudes. The results are also very useful for microphysical studies such as rain/snow growth mechanisms that require a vertical motions as a key input. The authors provide a good summary of past work and the manuscript in general is well written except for numerous typographical errors and some poorly worded sentences. The technical details are sufficient for the material presented. There is too much detail in some sections, and only the key points should be included (e.g., section 4.3).

Answer:

We appreciate the reviewer's comment. Actually, the Editor had pointed out the typographical errors after we submitted the original manuscript, then we sent the manuscript out for editorial service and submitted a revised version. However, when dealing with the technical comments, we found many typographical errors do exit in the old version, but have been corrected in the revised version. Maybe the reviewers were reading the old version. The revised version can be downloaded on http://www.atmos-chem-phys-discuss.net/acp-2015-1021/#discussion . Nevertheless, the sciences are the same. We have addressed the comments raised by the reviewer, the sample issue and limitations of aircraft measurements have been highlighted. Sections with too much detail are simplified. A discussion section has been added to show the complicated interactions among

vertical velocity, entrainment/detrainment and microphysics. We have changed the manuscript title to "Characteristics of Vertical Air Motion in Isolated Convective Clouds" to highlight that this study deals with isolated convections rather than mesoscale convective systems.

The paper deals with what shallow to moderate convection. The authors need to add some discussion in the abstract and conclusions on the fact that the measurements presented are still a biased sample of convection. Are the measurements truly representative of all convection in the three regions presented, or did for example the planes used stay away from stronger, and/or deeper convection, or ones with higher reflectivity. What the paper points out is that there are some similarities between the regions, but that there is really a wide variety of convective types over the globe. This is a good point to make in the paper that there are few measurements of this sort so they are greatly needed, but they represent specific regions and types of convection, more from other regions are needed, and one should not interpret the results that these results can be generalized globally. A few summary sentences (abstract, intro, and conclusions) on this point would make the paper better.

Answer:

We appreciate the comment. We totally agree that this study only deals with a biased sample of convective clouds. Only three field campaigns are analyzed and MCSs were not sampled. The results cannot be generalized globally. We have pointed this out in the revised manuscript, included in abstract, introduction, datasets description and conclusion. We also changed the manuscript title to "Characteristics of Vertical Air Motion in Isolated Convective Clouds" to highlight that this study deals with isolated convections rather than mesoscale convective systems.

In addition, we have added more text to point out the limitations of aircraft measurements. First, aircraft cannot provide 3-D information of the cloud, so the air mass flux is derived from measurements in single-line penetrations. Second, aircraft might not penetrate through the strongest part of drafts due to safety issues. Moreover, in-situ measurements only provide data from single-line penetrations, but the vertical velocities are very different at different heights in a cloud. For example, many penetrations in COPE are near cloud top, while in HiCu and ICE-T there are many penetrations far below cloud top. Therefore, readers need to be aware of the limitations of aircraft measurements when using the results in this study.

While I find the paper quite interesting, there could be more connection between the convection dynamics and microphysics. Processes such as mixing are barely mentioned in the text. It would be interesting for example to make connections between the updraft characteristics such as mass fluxes, diameters, and entrainment.

Answer:

We appreciate the comment. We have tried to explore the interactions between dynamics and microphysics, but the physical processes are very complicated and there are many limitations of aircraft instruments (e.g. resolution, time response and uncertainty) and sample issues. An example is given in Fig. R1. In the figure, we plot the mean vertical velocity (a and b), normalized relative humidity (c and d), normalized FSSP concentration and normalized King LWC (e and f) as a function of normalized scale from cloud edge to location of the maximum vertical velocity in the updraft closest to the cloud edge. On the x-axis, 0 indicates the cloud edge, 1 indicates the location of maximum vertical velocity in the updraft closest to the cloud edge, where is less affected by entrainment. As shown in the figure, weaker updraft associates with

lower relative humidity, droplet concentration and LWC, and stronger updraft associates with higher relative humidity, droplet concentration and LWC. This maybe partly due to entrainment/detrainment mixing. This figure is from ICE-T only because in HiCu and COPE we do not have fast-response instrument to measure RH. The droplet concentration and LWC may have large uncertainty because FSSP often has shattering issues and King probe cannot detect large drops (> 50um).

[Figure]

Fig. R1: Mean vertical velocity (a and b), normalized relative humidity (c and d), normalized FSSP concentration and normalized King LWC (e and f) as a function of normalized scale from cloud edge to updraft closest to the edge. 0 on the x-axis indicates the cloud edge, 1 on the x-axis indicates the location of maximum vertical velocity in the updraft closest to the cloud edge.

We also tried to use indirect ways to explore the impacts of entrainment on vertical velocity. Fig. R2 shows the PDF of vertical velocity in downdrafts near cloud edge and inside cloud. In HiCu and COPE the downdrafts near cloud edge are stronger than those inside clouds, maybe because of the strong evaporation-cooling effect induced by entrainment, while in ICE-T the downdrafts are similar near cloud edge and inside cloud. This only partly explains the stronger downdraft in HiCu and COPE than ICE-T, because the downdrafts inside clouds are also stronger in HiCu and COPE than ICE-T.

[Figure]

Fig. R2: PDFs of vertical velocity in downdrafts near cloud edge and inside cloud.

Due to the complexity of dynamics-microphysics interactions and the limitations of aircraft measurements, it is better to address this problem in detail in other papers. We have written a separated paper and discussed the interaction between vertical velocity and liquid-ice mass

partitioning (Yang et al. manuscript submitted to JAS), in which an algorithms is developed to partitioning liquid and ice mass using multiple in-situ instruments. An example is given in Fig. R3, the figure shows in developing cloud the LWC and IWC are higher in stronger updraft, but the liquid fraction has no obvious correlation with vertical velocity. In mature clouds, LWC is higher in stronger updrafts, but IWC is similar in weak and strong updrafts. Between -3 C and -8 C, the liquid fraction is smaller in weaker updrafts, maybe because secondary ice production (e.g. H-M process) is more significant in weaker updraft (Heymsfield and Willis 2014). Only ICE-T is used in that paper because in COPE and HiCu we do not have the appropriate instruments to provide sufficient measurements.

[Figure]

Fig. R3: The mean profiles of LWC, the IWC, and the liquid fraction as a function of temperature for the (a-c) young turrets and (d-f) mature turrets with vertical velocities of 1 m s$^{-1}$ – 4 m s$^{-1}$ (green), 4 m s$^{-1}$ – 7 m s$^{-1}$ (blue) and greater than 7 m s$^{-1}$ (purple).

In the revised manuscript, we decide to add a discussion section to highlight the importance of the interactions between dynamics and microphysics, and discuss the possible impacts of entrainment and microphysics on vertical velocity.

Technical and other details:

Lines 128-138: What are typical reflectivity in the convection. I know this will be from the W-band radar but this information would still be useful.

Answer:

The reflectivity depends on the stage of the clouds. The reflectivity in convective core is typically 10-20 dBZ in ICE-T, 5-20 dBZ in COPE, and 0-15 dBZ in HiCu. These reflectivity values may not reveal the maximum reflectivity in convective cores due to sampling issue. We've added this information in the text.

Lines 171: Define strong updrafts since these are still relatively weak compared to deeper convection.

Answer: We have changed "strong updrafts" to "relatively strong updrafts". And have added *"These drafts maybe strong for isolated convections, but not necessary strong compared to MSCs"*.

Line 189: This accuracy (0.2 m/s) is quite good. Is there any chance there are biases rather than random errors on the vertical velocity?

Answer: There are no other instruments as references to provide systematic errors on vertical velocity. In the three datasets we do not see unrealistic values of vertical velocity (except a few cases in which the instrument was not working, which have been excluded in the study). Generally, the 0.2 m/s could be seen as the systematic error, random error could be larger than 0.2 m/s.

Line 209: Footnote. This is roughly the sensitivity of CloudSat. Is this the reason -30 dBZ was chosen since there will be cloud at lower reflectivities?

Answer: We choose this threshold by plotting the reflectivity near flight level in cloud free air. As shown in Fig. 2 in the manuscript, the reflectivity near flight levels is about -30 dBZ in cloud free air due to WCR signal noise. At levels far above or below flight level, the noise level is higher. In this study we mainly use in-situ measurement, so we only consider the reflectivity near flight level. Clouds with reflectivity lower than the noise level cannot be identified by WCR, and are excluded in this study, most of them maybe not convective clouds.

Lines 233-235: How do you know the 2D symmetry of the updraft since you might not fly through the peak up and downdrafts? Both the W-band radar and in situ measurements will not tell you this.

Answer: Here we want to show whirling penetrations and penetrations with significant turns have been rejected, so the cloud scale will not be significantly overestimated. We have modified this sentence to make it clear.

Line 236: "there is no" should be "there are no"

Answer: The comment has been addressed in the revised manuscript.

Line 238: Excluding MCS biases the results. This comes down to emphasis in this paper on small to moderate convection rather than deep convection in MCSs. Might mention this to keep the scope of your study in perspective.

Answer: We have pointed out the sample issue in the revised manuscript, including abstract, introduction, datasets description and conclusion. We also changed the manuscript title to "Characteristics of Vertical Air Motion in Isolated Convective Clouds" to highlight that this study deals with isolated convections rather than mesoscale convective systems.

Figure 4: need labels for field experiment associated with each color.

Answer: Labels have been added.

Lines 264-268: It might be useful to plot one example of a trace through one of the updraft/downdraft penetrations. This would be helpful to understand some of the averaging performed.

Answer:

Good suggestion. But the clouds were randomly sampled in the three field campaigns, we do not have continuous penetrations in one updraft/downdraft. More data are needed in the future. In addition, in-situ data itself is not enough to resolve the fine structure, in Fig. 2 in the manuscript, we can see many fine structures from the Doppler velocity measured by WCR, in-situ measurements can capture the details at single levels.

Line 268: "turbulences" to "turbulence"

Answer: "turbulences" has been changed to "turbulence" in the revised manuscript.

Line 300: "convections" to "convection"

Answer: "convections" has been changed to "convection" in the revised manuscript.

Line 294: "strong draft" – I would again put this in perspective since it is strong in your study, but not necessarily strong with respect to MCS updrafts for example.

Answer: We have add a sentence to indicate the definition of "strong" is only for this study, but not necessarily strong with respect to other convections (e.g. MCS): *"The definition of "weak", "moderate" and "strong" only apply for this study. Other convections (e.g. MCS) could have much stronger updrafts."*

Line 323-324: Again, this point should be more prominent in the paper.

Answer: We have highlighted that the definition of "weak", "moderate" and "strong" only apply for this study.

Lines 386-387: This statement should be in the summary/conclusions since these measurements are important but we need a lot more.

Answer: We have added this statement in the conclusion.

Line 397: "results" to "result"

Answer: "results" has been changed to "result" in the revised manuscript.

Line 403: "relatively" to "relative"

Answer: "relatively" has been changed to "relative" in the revised manuscript.

Figure 10 and similar plots: I find these plots a little complicated and confusing but probably acceptable for publication. I can't think of an alternative but possibly there is a better way to plot.

Answer: We tried to improve the figures but haven't found a better way, because there are a lot of information in the figure. We have modified the text to better describe this figure.

Line 422: Some of the statements in this section should go in a summary and conclusions section. Points like lines 468-473 are important summary statements. You should consider pulling some of the summary points like this and putting them in the conclusions.

Answer: Statements with key points have been added in the conclusion.

Line 469: "pervious" to "previous"

Answer: "pervious" has been changed to "previous" in the revised manuscript.

Lines 509-510: "While in this study" should be something like "In contrast, this study shows the strongest. . .."

Answer: The sentence has been changed to *"In contrast, this study shows the strongest updrafts and downdrafts were observed at higher levels"*.

Line 522: "When exclude" to "When we exclude" Line 536: "convective cloud" to "convective clouds"

Answer: The comment has been addressed in the revised manuscript.

Lines 539-540: Can you say anything about the two-dimensionality of drafts from the remote sensing data?

Answer: We have added the following sentence in the text: "for example, airborne radar with slant and zenith/nadir viewing beams can provide two-dimensional wind structure in convective clouds".

Line 553: "with expectation" to "expected"

Answer: "with expectation" has been changed to "expected" in the revised manuscript.

Line 564: "Since . . ..to better". This is an obvious fact. May want to just say that the aircraft just provides a line of data through drafts, and not vertical information unless the plane makes multiple passes through the same cell.

Answer: We have changed to sentence to *"Since the aircraft just provides a line of data through drafts, and not vertical information unless the plane makes multiple passes through the same cell, more data, including remote sensing measurements are needed to better understand the evolution of the vertical velocity in convective clouds at different stages."*

Line 568: in the Summary section, you should reiterate the criteria for considering up/downdrafts, i.e., >xx m/s.

Answer: The criteria for considering up/downdrafts is reiterated.

Line 596: Flux calculations assume two-dimensionality of drafts and this might not be the case. Should mention this as a weakness in the study, i.e., using a single line penetration through drafts to make flux calculations.

Answer: We have highlighted that due to the limitation of aircraft measurements, the air mass flux is calculated using the data from single line penetrations. This may not fully capture the real air mass flux in the clouds and is a weakness of this study.

---

## Author Comment (AC2) · 23 May 2016

Reviewer's comments in black, replies in blue.

I always like to see an in-depth study of vertical motions in the atmosphere because, as the authors point out, understanding these is vital to improving our understanding of (and hence modeling capabilities) many processes influenced by vertical motions. First, before I get to the science, this document was not ready for submission in any form. It is riddled with typographical errors making it very difficult to get to the science. I started to list them but, frankly, this is the job of an editorial service, something I recommend the author take advantage of. For example: "The COPE project was conducted from 03 July to 21 August, 2013". This is not English.. "The COPE project was conducted from the 3rd of July to the 21st August, 2013".. Write in English not in code. I have two broad areas of concern with this manuscript:

Answer:

We appreciate the reviewer's comment and sorry for the typographical errors. Actually, the Editor had pointed out the typographical errors after we submitted the original manuscript, then we sent the manuscript out for editorial service and submitted a revised version. However, when dealing with the technical comments raised by Reviewer 1, we found that many typographical errors pointed out by the Reviewer 1 exit in the old version, but have been corrected in the revised version. Maybe the reviewers were reading the old version. The revised version can be downloaded on http://www.atmos-chem-phys-discuss.net/acp-2015-1021/#discussion . In this round of revision, we have corrected a few more typographical errors.

I have two broad areas of concern with this manuscript:

1) The authors do not address the idea of sample size or sample bias OR more importantly geometric issues of sampling, in a line, a 2/3D object (being an updraft core). See Giangrande et al 2013 for a discussion of issues with profiler systems and angle of attack. Basically if you dissect an updraft core how do you know if you hit the strongest part of the updraft? Furthermore, up until the end, the idea of selection bias is not addressed. Even the C-130 will avoid the strongest cores. You can not build a PDF out to the tail from aircraft measurements. You can, as the paper did somewhat, look at intrinsic updraft properties. But you can not look at the distribution. I am somewhat disappointed , given the brief reference to microphysical measurements, that the authors did not relate vertical motions to microphysical properties of the updraft cores. This is something in-situ platforms are uniquely capable of doing. Also, in the literature review of methodologies for measuring vertical motions the authors neglect scanning radar measurements such as those shown in Collis et al 2013 and Nicol et al 2015 (not to mention a raft of airborne radar measurements from the NOAA p3 (look for papers from Jorgensen) and other aircraft that use the vertical plus 45 degree tilt methods.

Answer:

We totally agree with the reviewer that there are many limitations in aircraft measurements. First, aircraft might not penetrate through the strongest part of drafts due to safety issues. In addition, aircraft cannot provide 3-D information of the cloud, and the air mass flux is derived from measurements in single-line penetrations. Moreover, this study only deals with isolated convective clouds. Only three field campaigns are analyzed and MCSs are excluded in this study. The results cannot be generalized globally. We have pointed out these weaknesses in the revised manuscript, including abstract, introduction, datasets description and conclusion. We also changed the manuscript title to "Characteristics of Vertical Air Motion in Isolated Convective

Clouds" to highlight that this study deals with isolated convections rather than mesoscale convective systems (MCSs).

For the PDF distributions, we think it will be good to keep them the paper even though there are potential sampling issues. First, modelers do need the aircraft measurements to provide PDF distributions of vertical velocities (personal communications: Guangjun Zhang, Xiaohong Liu and Sungsu Park). Second, due to the relative small sizes of isolated convective clouds, the sampling bias associated with where to penetrate clouds is not as large as sampling MCSs. During the sampling of isolated convective clouds, we typically aligned the central part of cloud to penetrate at the flight height. During ICE-T and COPE, we have penetrations in updrafts stronger than 20 m/s (please note this is just for isolated convections, in which the updrafts are weaker than MCSs), and previous studies based on in-situ data rarely reported such relatively strong updrafts. Actually, this is one of our motivations to make this study. The PDFs can also be used to evaluate and improve remote sensing retrievals because in-situ measurements are more accurate than remote sensing, especially in mixed-phase convective clouds. Then remote sensing can provide PDFs out to the tail. Therefore, the PDFs in the paper still provide valuable information, but readers do need to be aware of the weaknesses and limitations of aircraft measurements.

We tried to explore the interactions between microphysics and vertical velocity, but the physical processes are very complicated, and there are many limitation of aircraft instruments in measuring the microphysics in mixed-phase convective clouds. For example, FSSP has the shattering issue, hot-wire probes often underestimates the LWC because there are many large drops which cannot be directly sampled by these probes. Due to the complexity of dynamics-microphysics interactions and the limitations of aircraft measurements, it is better to

address this problem in detail in other papers. We have written a separated paper and discussed

the interaction between vertical velocity and liquid-ice mass partition in the mixed-phase cloud

region within convective clouds (Yang et al. manuscript submitted to JAS), in which an

algorithms is developed to partitioning liquid and ice mass using multiple in-situ instruments. An

example is given in Fig. R1, the figure shows in developing cloud the LWC and IWC are higher

in stronger updraft, but the liquid fraction has no obvious correlation with vertical velocity. In

mature clouds, LWC is higher in stronger updrafts, but IWC is similar in weak and strong

updrafts. Between -3 C and -8 C, the liquid fraction is smaller in weaker updrafts, maybe

because secondary ice production is more significant in weaker updraft (Heymsfield and Willis

2014), results in relatively larger fraction of IWC. Such in-depth analyses only can be applied to

ICE-T measurements in that paper because in COPE and HiCu we do not have the appropriate

instruments to provide sufficient measurements.

[Figure]

Fig. R1: The mean profiles of LWC, the IWC, and the liquid fraction as a function of temperature for the (a-c) young turrets and (d-f) mature turrets with vertical velocities of 1 m s$^{-1}$ – 4 m s$^{-1}$ (green), 4 m s$^{-1}$ – 7 m s$^{-1}$ (blue) and greater than 7 m s$^{-1}$ (purple).

Other than the interactions between vertical velocity and microphysics, entrainment/detrainment mixing also have impact on vertical velocity. But due to the complexity of the physical processes and the limitations of aircraft instruments, we think it is better to address this problem in detail in separated paper as well. (Please see the reply to Reviewer 1's comments).

In the revised manuscript, we add a discussion section to highlight the importance of the interactions between dynamics and microphysics, and discuss the possible impacts of entrainment and microphysics on vertical velocity.

In the revised manuscript, we have added the literatures about ground-based and airborne volumetric radar measurements in Introduction. For example, "*Collis et al. (2013) provides statistics of updraft velocities for difference convective cases near Darwin, Australia using retrievals from ground-based scanning Doppler radars and a multifrequency profiler*". "*Airborne volumetric Doppler radars have also been used to study the dynamic structure of convective clouds (e.g. Jorgensen and Smull 1993; Hildebrand et al. 1996; Jorgensen et al. 2000)*". "*Remote sensing has the advantage of being able to measure the vertically velocity at different heights simultaneously (Tonttila et al., 2011), and some of the techniques can detect the strongest updraft cores in convective clouds (Heymsfield et al. 2010; Collis et al. 2013)*". "*Volumetric radars can provide three-dimensional (3D) structure of air motion in convective clouds (Collis et al. 2013; Nicol et al. 2015; Jorgensen et al. 2000)*".

2) This comment relates to a specific question asked by the Journal in its review criteria "Are substantial conclusions reached?". I am deeply concerned by the authors attempt to relate the three field programs and say something about maritime versus continental convection. For one, the author did not put the cases into context. What was the CAPE for various cases? etc.. A selection of clouds at each campaign a climatology does not make. While the author caveats his comparison even the attempt to contrast the different regime is dangerous. For one, as mentioned, the strongest cores in the region of HiCu would all but destroy even the C-130 (See the various photos associated with the Byers et al study of hail damage). To attempt to make a comparison, then state it goes contrary to common conception (Continental » Maritime) and then turn around and say "we did not sample the strongest updrafts in the continental case" is disingenuous.

So negatives out of the way, one of the things that redeem the paper is the focus on updraft shape and how that varies with height. Personally I find this very interesting as not only does the mass flux of a plume influence transport but the vertical velocity within determines many microphysical aspects. ie a plume that starts thin and then expand for the same mass flux would have lower vertical velocities aloft influencing processes like Hallett-Mossop splintering etc.. (and associated latent feedbacks).. The paper should focus more on this and the *intrinsic* differences. Things that are co-varying and less susceptible to sampling and decision bias.

Answer:

We appreciate the reviewer's comments. We have pointed out the weaknesses of aircraft measurements in the revised manuscript, including abstract, introduction, datasets description and conclusion, as well as the title.

Due to the limitation of aircraft measurement, we have deleted some results which are sensitive to the sampling issue. For example, "the vertical velocity in HiCu is weaker than that in COPE and ICE-T". In addition, in this paper we plot the vertical velocity PDFs and profiles as a function of height MSL (Fig. 8 and 10), so at the same height, the vertical velocity maybe weaker in HiCu. However, the updrafts were strengthening with height, and some updrafts could be close to 20 m/s at > 6 km MSL (Fig. 8) in HiCu. Maybe at higher levels the updrafts in HiCu were stronger than COPE and ICE-T, but we do not have more data. If we plot the updraft PDFs and profiles as a function of height above cloud base, the results in HiCu maybe closer to that in COPE and ICE-T. However, cloud base heights are variable and we do not have data to calculate the cloud base heights.

In the revised paper, we have added some text to describe the ambient conditions which many affect the vertical air motion. For example, *"the convective available potential energy (CAPE) in ICE-T is greater than 2000 J kg$^{-1}$. The CAPE in COPE is typically a few hundred J kg$^{-1}$. No soundings are available for HiCu, so we have to use aircraft measurements to estimate the CAPE. In some cases, the full CAPE cannot be calculated since the aircraft only flew at low levels (< 10 km MSL). The aircraft measurements suggest the CAPE in HiCu ranges from less than 100 J kg$^{-1}$ to more than 500 J kg$^{-1}$"*.

As suggested by the reviewer, we have added more discussion about the *intrinsic* differences among the three field campaigns. For example, the downdrafts in HiCu and COPE are obviously stronger than that in ICE-T, maybe partly due to the evaporation-cooling effect induced by entrainment (please see the reply to Reviewer 1). We also changed Fig. 11 to Fig. R2 as follows to show how the draft shape changes with height. Actually, the evolution of draft with height is very complicated. Based on our datasets, there could be different possibilities: 1) an

updraft expands and the vertical velocity weakens with height, 2) an updraft expands and the vertical velocity strengthens with height, 3) an updraft splits to multiple updrafts and downdrafts, 4) two updrafts merged and become one updrafts. Since we do not have continuous penetrations in a single cloud, we have to statistically analyze the evolution of draft shape. In Fig. R2, we can see that the normalized shape do not have significantly change with height, the peak vertical velocity is strengthening with height. Connecting this figure to diameter (Fig. 4), vertical velocity (Fig. 8) and air mass flux (Fig. 9), the results show statistically, the drafts were expanding (Fig. 4) and the vertical velocity was strengthening (Fig. R2 and 8), but the air mass flux was not increasing (Fig. 9). This reveals the complicated physical processes (e.g. entrainment, water loading and the possibilities described above). The interaction between vertical velocity evolution and microphysics is even more complicated and needs to be analyzed in detail in separated papers (please see the reply to the first comment above).

[Figure]

Fig. R2: Composite structure of the vertical velocity as a function of the normalized diameter for the updrafts and downdrafts with air mass flux $\geq 10$ kg m$^{-1}$ s$^{-1}$ in magnitude. The 0 and 1 coordinates on the x-axis indicate the upwind and downwind sides of the draft.

Finally, we want to say this paper is just a part of the whole picture. The physical processes in mixed-phase convective clouds (e.g. interaction between dynamics and microphysics) are very complicated, and need to be further explored in the future with more experimental data, especially with more advanced measurements. The contributions of this paper are 1) provides statistical results of vertical air motion in isolated convective clouds using in-situ data in recent

field campaigns, which could be used to evaluate remote sensing retrievals and model simulations. 2) In-situ measurements of vertical velocity stronger than 20 m/s in isolated convective clouds are provided. Previous studies using in-situ measurement rarely had penetrations in such relatively strong updrafts. 3) This paper highlights the importance of small drafts using high-resolution in-situ data, which is not shown in previous studies. 4) Some 'intrinsic' differences and similarities of vertical air motions among the three field campaigns are discussed. Aircraft measurements do have many limitations and this paper only deals with isolated convections, we have highlighted them in the revised paper.

---

## Author Response (AR2)

Editor and reviewer's comments in black, replies in blue.

Comments from Editor:

The reviewers are mostly satisfied with the technical aspects of the paper, but there are clearly some remaining presentation issues. Please go through the entire paper carefully and edit it for grammar, word-usage, clarity, etc., including but not limited to those issues identified by the reviewers.

The next iteration will be an editor-only review, i.e. it will not go back to the reviewers.

Answer:

We appreciate the comment. We have gone through the manuscript carefully and addressed the presentation issues.

Comments from Reviewer 2:

The paper is significantly improved. The authors have answered my concerns well although I do have some concern at the statement : " First, modelers do need the aircraft measurements to provide PDF distributions of vertical velocities (personal communications: Guangjun Zhang, Xiaohong Liu and Sungsu Park). " I completely agree modellers NEED pdfs of updrafts they need *correct* PDFs OR they need to know the shortfalls and applicability of these observations so as to correctly validate output from GCMs. The last thing we want to be doing as observationalists is to be introducing new biases into models by using unrepresentative data.

To this end I would suggest a final pass of the paper and where you have a statement like on line 22: "The downdrafts are stronger ..." I suggest the insertion of "observed" so The observed downdrafts are stronger". This caveats the dataset against observational issues. You do not know if all the updrafts in ICE-T were stronger than COPE all you know is what you observe.

The manuscript still contains many typographical errors, mainly in the corrections added. The Authors need to get a final edit done before submission

All in all the manuscript is now more honest to the data and is nearing a standard acceptable for publication.

Answer:

We appreciate the comment. We agree modelers need to know the shortfalls and applicability of observations, which have been pointed out in the manuscript.

We have gone through the manuscript carefully and addressed the presentation issues.

[revised manuscript text omitted]